# THE ASYMMETRIC MAXIMUM MARGIN BIAS OF QUASI-HOMOGENEOUS NEURAL NETWORKS

**Daniel Kunin**[*] **& Atsushi Yamamura** (山村篤志)[*]
Stanford University
{kunin,atsushi3}@stanford.edu

**Chao Ma, Surya Ganguli**
Stanford University
{chaoma,sganguli}@stanford.edu

## ABSTRACT

In this work, we explore the maximum-margin bias of *quasi-homogeneous* neural networks trained with gradient flow on an exponential loss and past a point of separability. We introduce the class of quasi-homogeneous models, which is expressive enough to describe nearly all neural networks with homogeneous activations, even those with biases, residual connections, and normalization layers, while structured enough to enable geometric analysis of its gradient dynamics. Using this analysis, we generalize the existing results of maximum-margin bias for homogeneous networks to this richer class of models. We find that gradient flow implicitly favors a subset of the parameters, unlike in the case of a homogeneous model where all parameters are treated equally. We demonstrate through simple examples how this strong favoritism toward minimizing an asymmetric norm can degrade the *robustness* of quasi-homogeneous models. On the other hand, we conjecture that this norm-minimization discards, when possible, unnecessary higher-rate parameters, reducing the model to a sparser parameterization. Lastly, by applying our theorem to sufficiently expressive neural networks with normalization layers, we reveal a universal mechanism behind the empirical phenomenon of *Neural Collapse*.

## 1 INTRODUCTION

Modern neural networks trained with (stochastic) gradient descent generalize remarkably well despite being trained well past the point at which they interpolate the training data and despite having the functional capacity to memorize random labels Zhang et al. (2021). This apparent paradox has led to the hypothesis that there must exist an implicit process biasing the network to learn a "good" generalizing solution, when one exists, rather than one of the many more "bad" interpolating ones. While much research has been devoted to identifying the origin of this *implicit bias*, much of the theory is developed for models that are far simpler than modern neural networks. In this work, we extend and generalize a long line of literature studying the *maximum-margin bias* of gradient descent in *quasi-homogeneous networks*, a class of models we define that encompasses nearly all modern feedforward neural network architectures. Quasi-homogeneous networks include feedforward networks with homogeneous nonlinearities, bias parameters, residual connections, pooling layers, and normalization layers. For example, the ResNet-18 convolutional network introduced by He et al. (2016) is quasi-homogeneous. We prove that after surpassing a certain threshold in training, gradient flow on an exponential loss, such as cross-entropy, drives the network to a maximum-margin solution under a norm constraint on the parameters. Our work is a direct generalization of the results discussed for homogeneous networks in Lyu & Li (2019). However, unlike in the homogeneous setting, the norm constraint only involves a subset of the parameters. For example, in the case of a ResNet-18 network, only the last layer's weight and bias parameters are constrained. This asymmetric norm can have non-trivial implications on the robustness and optimization of quasi-homogeneous models, which we explore in sections 5 and 6.

---

[*]Equal contribution. Correspondence to Daniel Kunin and Atsushi Yamamura.

## 2 BACKGROUND AND RELATED WORK

Early works studying the maximum-margin bias of gradient descent focused on the simple, yet insightful, setting of logistic regression Rosset et al. (2003); Soudry et al. (2018). Consider a binary classification problem with a linearly separable[1] training dataset $\{x_i, y_i\}$ where $x_i \in \mathbb{R}^d$ and $y_i \in \{-1, 1\}$, a linear model $f(x; \beta) = \beta^\intercal x$, and the exponential loss $\mathcal{L}(\beta) = \sum_i e^{-y_i f(x_i; \beta)}$. As shown in Soudry et al. (2018), the loss only has a minimum in $\beta$ as its norm becomes infinite. Thus, even after the network correctly classifies the training data, gradient descent decreases the loss by forcing the norm of $\beta$ to grow in an unbounded manner, yielding a slow alignment of $\beta$ in the direction of the maximum $\ell_2$-margin solution, which is the configuration of $\beta$ that minimizes $\|\beta\|$ while keeping the margin $\min_i y_i f(x_i; \beta)$ at least 1. But what if we parameterize the regression coefficients differently? As shown in Fig. 1, different parameterizations, while not changing the space of learnable functions, can lead to classifiers with very different properties.

**Linear networks.** An early line of works exploring the influence of the parameterization on the maximum-margin bias studied the same setting as logistic regression, but where the regression coefficients $\beta$ are multilinear functions of parameters $\theta$. Ji & Telgarsky (2018) showed that for deep linear networks, $\beta = \prod_i W_i$, the weight matrices asymptotically align to a rank-1 matrix, while their product converges to the maximum $\ell_2$-margin solution. Gunasekar et al. (2018) showed that linear diagonal networks, $\beta = w_1 \odot \cdots \odot w_D$, converge to the maximum $\ell_{2/D}$-margin solution, demonstrating that increasing depth drives the network to sparser solutions. They also show an analogous result holds in the frequency domain for full-width linear convolutional networks. Many other works have advanced this line of literature, expanding to settings where the data is not linearly separable Ji & Telgarsky (2019), generalizing the analysis to other loss functions with exponential tails Nacson et al. (2019b), considering the effect of randomness introduced by stochastic gradient descent Nacson et al. (2019c), and unifying these results under a tensor formulation Yun et al. (2020).

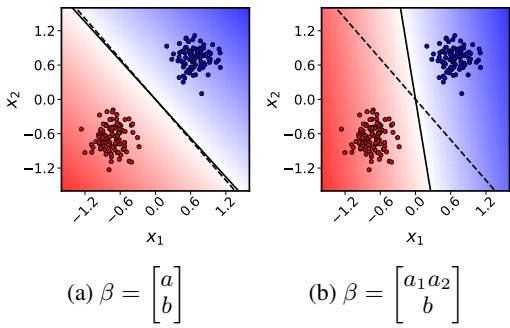

(a) $\beta = \begin{bmatrix} a \\ b \end{bmatrix}$      (b) $\beta = \begin{bmatrix} a_1 a_2 \\ b \end{bmatrix}$

Figure 1: **Maximum-margin bias changes with parameterization.** Logistic regression, $f(x) = \beta^\intercal x$, trained with gradient descent on a homogeneous (left) and quasi-homogeneous (right) parameterization of the regression coefficients $\beta$. The dashed black line is the maximum $\ell_2$-margin solution and the solid black line is the gradient descent trained classifier after $1e5$ steps. Existing theory predicts the homogeneous model will converge to the maximum $\ell_2$-margin solution. In this work we will show that the quasi-homogeneous model is driven by a different maximum-margin problem.

**Homogeneous networks.** While linear networks allowed for simple and interpretable analysis of the implicit bias in both the space of $\theta$ (*parameter space*) and the space of $\beta$ (*function space*), it is unclear how these results on linear networks relate to the behavior of highly non-linear networks used in practice. Wei et al. (2019) and Xu et al. (2021) made progress towards analysis of non-linear networks by considering shallow, one or two layer, networks with *positive-homogeneous* activations, i.e., there exists $L \in \mathbb{R}_+$ such that $f(\alpha x) = \alpha^L f(x)$ for all $\alpha \in \mathbb{R}_+$. More recently, two concurrent works generalized this idea by expanding their analysis to all positive-homogeneous networks. Nacson et al. (2019a) used vanishing regularization to show that as long as the training error converges to zero and the parameters converge in direction, then the rescaled parameters of a homogeneous model converges to a first-order Karsh-Kuhn-Tucker (KKT) point of a maximum-margin optimization problem. Lyu & Li (2019) defined a normalized margin and showed that once the training loss drops below a certain threshold, a smoothed version of the normalized margin monotonically converges, allowing them to conclude that all rescaled limit points of the normalized parameters are first-order KKT points of the same optimization problem. A follow up work, Ji & Telgarsky (2020), developed a theory of unbounded, nonsmooth Kurdyka-Lojasiewicz inequalities to prove a stronger result of directional convergence of the parameters and alignment of the gradient with the parameters along the gradient flow path. Lyu & Li (2019) and Ji & Telgarsky (2020) also explored empirically non-homogeneous models with bias parameters and Nacson et al. (2019a) considered

---

[1]Linearly separable implies there exists a $w \in \mathbb{R}^d$ such that for all $i \in [n]$, $y_i w^\intercal x_i \geq 1$.

theoretically non-homogeneous models defined as an ensemble of homogeneous models of different orders. While these works have significantly narrowed the gap between theory and practice, all three works have also highlighted the limitation in applying their analysis to architectures with bias parameters, residual connections, and normalization layers, a limitation we alleviate in this work. In a parallel literature studying low-rank biases in deep learning, Le & Jegelka (2022) analyzed non-homogeneous models where the nonlinearities are restricted to the first few layers.

## 3 DEFINING THE CLASS OF QUASI-HOMOGENEOUS MODELS

Here we introduce the class of quasi-homogeneous models, which is expressive enough to describe nearly all neural networks with positive-homogeneous activations, while structured enough to enable geometric analysis of its gradient dynamics. Throughout this work, we will consider a binary classifier $f(x; \theta) : \mathbb{R}^d \to \mathbb{R}$, where $\theta \in \mathbb{R}^m$ is the vector concatenating all the parameters of the model. We assume the dynamics of $\theta(t)$ over time $t$ are governed by gradient flow $\frac{d\theta}{dt} = -\frac{\partial \mathcal{L}}{\partial \theta}$ on an exponential loss $\mathcal{L}(\theta) = \frac{1}{n} \sum_i e^{-y_i f(x_i; \theta)}$ computed over a training dataset $\{(x_1, y_1), \ldots, (x_n, y_n)\}$ of size $n$ where $x_i \in \mathbb{R}^d$ and $y_i \in \{-1, 1\}$. In App. H we generalize our results to multi-class classification with the cross-entropy loss.

**Definition 3.1** ($\Lambda$-Quasi-Homogeneous). *For a (non-zero) positive semi-definite matrix $\Lambda \in \mathbb{R}^{m \times m}$, a model $f(x; \theta)$ is $\Lambda$-quasi-homogeneous if under the parameter transformation*

$$\psi_\alpha(\theta) := e^{\alpha \Lambda} \theta, \tag{1}$$

*the output of the model scales $f(x; \psi_\alpha(\theta)) = e^\alpha f(x; \theta)$ for all $\alpha \in \mathbb{R}$ and input $x$.*

In this work, we assume $\Lambda$ is diagonal[2] and let $\lambda_i = (\Lambda)_{ii}$ and $\lambda_{\max} = \max_i \lambda_i$ be the maximum diagonal element, which must be positive. Definition (3.1) generalizes the notion of positive homogeneous functions, allowing different scaling rates for different parameters to yield the same scaling of the output. Given two parameters with different values of $\lambda$, we refer to the parameter with larger $\lambda$ as *higher-rate* and the other as *lower-rate*.

**Examples.** We consider some simple quasi-homogeneous networks that are not homogeneous.

*Unbalanced linear diagonal network.* Consider a diagonal network as described in Gunasekar et al. (2018), but with a varying depth for different dimensions of the data. The regression coefficient $\beta_i$ for input component $x_i$ is parameterized as the product of $D_i \in \mathbb{N}$ parameters, yielding $f(x; \theta) = \sum_i (\prod_{j=1}^{D_i} \theta_{ij}) x_i$. When the $D_i$ are equal, the network is homogeneous, otherwise, the network is quasi-homogeneous where the choice of $\lambda$ can be $D_i^{-1}$ for $\theta_{ij}$.

*Fully connected network with biases.* One of the simplest quasi-homogeneous models is a multi-layer, fully-connected network with bias parameters, such as the two-layer network, $f(x; \theta) = w^2 \sigma \left( \sum_i w_i^1 x_i + b^1 \right) + b^2$, where $\sigma(\cdot)$ is a Rectified Linear Unit (ReLU). Without biases this network would be homogeneous, but their inclusion requires a quasi-homogeneous scaling of parameters to uniformly scale the output of the model. For example, the choice of $\lambda$ can be 1 for $b^2$ and $1/2$ for all other parameters.

*Networks with residual connections.* Similar to networks with biases, residual connections result in a computational path that requires a quasi-homogeneous scaling of the parameters. For example, the model $f(x; \theta) = \sum_j w_j^2 \sigma \left( \sum_i w_{ji}^1 x_i + x_j \right)$ is quasi-homogeneous, where the choice of $\lambda$ can be 1 for $w^2$ and 0 for $w^1$.

*Networks with normalization layers.* As discussed in Kunin et al. (2020), when normalization layers, such as batch normalization, are introduced into a homogeneous network, they induce scale invariance in the parameters in the preceding layer. However, as long as the last layer is positive homogeneous, then a network with normalization layers is quasi-homogeneous. For example, the network $f(x; \theta) = \sum_i w_i h_i(\theta', x) + b$ is quasi-homogeneous, where $w$ is the weight of the last layer, $b$ is the bias, $\theta'$ is the set of parameters in earlier layers, and $h(\theta', x)$ is the activation of the last hidden layer after normalization. The choice of $\lambda$ can be 1 for $w$ and $b$ and 0 for $\theta'$.

See App. A for more examples of quasi-homogeneous models and their relationship to ensembles of homogeneous networks of different orders, as discussed in Nacson et al. (2019a).

---

[2]When $\Lambda$ is not diagonal, by reparameterizing the model $\theta \to O\theta$ with a proper orthogonal matrix $O$, we can diagonalize $\Lambda$.

**Geometric properties.** Like homogeneous functions, quasi-homogeneous functions have certain geometric properties of their derivatives. Analogous to *Euler's Homogeneous Function Theorem*, for a quasi-homogeneous $f(x; \theta)$, we have $\langle \nabla_\theta f(x; \theta), \Lambda\theta \rangle = f(x; \theta)$, which is easily derived by evaluating the derivative $\nabla_\alpha f(x; \psi_\alpha(\theta))$ at $\alpha = 0$, the identity element of the transformation. Analogous to how the derivative of a homogeneous function of order $L$ is a homogeneous function of order $L-1$, the derivative of a quasi-homogeneous function under the same transformation respects the following property, $\nabla_\theta f(x; \psi_\alpha(\theta)) = e^{\alpha(I-\Lambda)} \nabla_\theta f(x; \theta)$. See App. A for a derivation of the geometric properties of quasi-homogeneous functions.

**Characteristic curves.** Throughout this work we consider the partition of parameter space into the family of one-dimensional *characteristic curves* mapped out by the parameter transformation in Eq. 1. The vector field generating the transformation, $\frac{\partial \psi_\alpha}{\partial \alpha}\big|_{\alpha=0} = \Lambda\theta$, is tangent to the characteristic curve and thus we will refer to this vector as the *tangent vector*. We define the *angle* $\omega$ between the *velocity* $\frac{d\theta}{dt}$ and tangent vector such that the *cosine similarity* between these two vectors is $\beta := \cos(\omega) = \frac{\langle \Lambda\theta, \frac{d\theta}{dt} \rangle}{\|\Lambda\theta\| \|\frac{d\theta}{dt}\|}$.

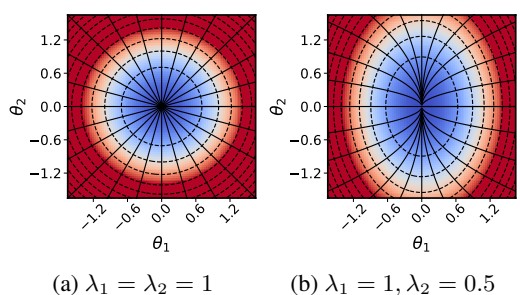

(a) $\lambda_1 = \lambda_2 = 1$      (b) $\lambda_1 = 1, \lambda_2 = 0.5$

Figure 2: **A natural coordinate system for quasi-homogeneous models.** A useful coordinate system for studying the gradient dynamics of quasi-homogeneous models is the decomposition of parameter space into characteristic curves (solid lines) and level sets of the $\Lambda$-seminorm (dashed lines). For a homogeneous function (left), this decomposition is equivalent to a polar decomposition. For a quasi-homogeneous function (right), then the directions of the characteristic curves are eventually dominated by the highest-rate parameters and the level sets of the $\Lambda$-seminorm are concentric ellipsoids.

**$\Lambda$-Seminorm.** The characteristic curves perpendicularly intersect a family of concentric ellipsoids defined by the $\Lambda$-*seminorm*, $\|\theta\|_\Lambda^2 := \sum_i \lambda_i \theta_i^2$. Together, the intersection of a given characteristic curve with an ellipsoid of given $\Lambda$-seminorm uniquely defines a single point in parameter space. In the setting of homogeneous networks, this geometric structure is equivalent to a polar decomposition of parameter space. We also define the $\Lambda$-normalized parameters $\hat{\theta} = \psi_{-\tau}(\theta)$ where $\tau(\theta)$ is implicitly defined such that $\|\hat{\theta}\|_\Lambda^2 = 1$. This corresponds to a unique projection of parameter $\theta$ onto the unit $\Lambda$-seminorm ellipsoid by moving along a characteristic curve.

As shown in Fig. 2, for a homogeneous function, the characteristics are rays and the $\Lambda$-seminorm is proportional to the Euclidean norm $\|\theta\|$. For a quasi-homogeneous function, then the directions of the characteristic curves and the $\Lambda$-seminorm are eventually dominated by the highest-rate parameters. Thus, we will also find it helpful to define the $\Lambda_{\max}$-*seminorm* as $\|\theta\|_{\Lambda_{\max}}^2 := \sum_{i:\lambda_i = \lambda_{\max}} \lambda_i \theta_i^2$.

## 4 QUASI-HOMOGENEOUS MAXIMUM-MARGIN BIAS

Having defined the class of quasi-homogeneous models and identified a natural coordinate system to explore their gradient dynamics, we now generalize the maximum-margin bias theory developed in Lyu & Li (2019) for homogeneous models to a general quasi-homogeneous model $f(x; \theta)$. Following the analysis strategy of Lyu & Li (2019), we make the following assumptions:

- **A1** (*Quasi-Homogeneous*). There exists a non-zero diagonal positive semi-definite matrix $\Lambda$, such that the model $f(x; \theta)$ is $\Lambda$-quasi-homogeneous.
- **A2** (*Regularity*). For any fixed $x$, $f(x; \theta)$ is locally Lipschitz and admits a chain rule[3].
- **A3** (*Exponential Loss*). $\mathcal{L}(\theta) = \frac{1}{n} \sum_i \ell_i$ where $\ell_i = e^{-y_i f(x_i; \theta)}$.
- **A4** (*Gradient Flow*). Learning dynamics are governed by $\frac{d\theta}{dt} \in \partial_\theta^\circ \mathcal{L}$ [4] for all $t > 0$.
- **A5** (*Strong Separability*). There exists a time $t_0$ such that $\mathcal{L}(\theta(t_0)) < n^{-1}$.

We also make the following additional assumptions not presented in Lyu & Li (2019):

---

[3]Nearly all neural networks have this property, including those with ReLU activations. For details, see Davis et al. (2020) or Lyu & Li (2019).

[4]The Clarke's subdifferential $\partial_\theta^\circ \mathcal{L}$ is a generalization of $\nabla_\theta \mathcal{L}$ for locally Lipschitz functions. For details, see App. A

- **A6** (*Normalized Convergence*). $\lim_{t \to \infty} \hat{\theta}(t)$ exists.
- **A7** (*Conditional Separability*). There exists a $\kappa > 0$ such that only $\theta$ with $\|\hat{\theta}\|_{\Lambda_{\max}} \geq \kappa$ can separate the training data.

A6 implies the convergence of the decision boundary and A7 implies that $\lambda_{\max}$ parameters play a role in the classification task. A6 is necessary for a technical reason, but we expect that this assumption can be weakened by exploiting the argument in Ji & Telgarsky (2020). A7 is trivially true for a homogeneous model where $\|\hat{\theta}\|_{\Lambda_{\max}} = \|\hat{\theta}\| = 1$, but not for a quasi-homogeneous model. In section 5 we will consider what happens when we remove this assumption. We now state our main theoretical result:

**Theorem 4.1** (Quasi-Homogeneous Maximum-Margin). *Under assumptions A1 to A7, there exists an $\alpha \in \mathbb{R}$ such that $\psi_\alpha(\lim_{t \to \infty} \hat{\theta}(t))$ is a first-order KKT point[5] of the optimization problem:*

$$
\begin{aligned}
&\textit{minimize} && \frac{1}{2}\|\theta\|_{\Lambda_{\max}}^2 \\
&\textit{subject to} && y_i f(x_i; \theta) \geq 1 \quad \forall i \in [n]
\end{aligned}
\tag{P}
$$

**Significance.** Theorem 4.1 implies that after interpolating the training data, the learning dynamics of the model are driven by a competition between maximizing the margin in function space and minimizing the $\Lambda_{\max}$-seminorm in parameter space. At first glance, this might seem like a straightforward generalization of the result discussed in Lyu & Li (2019) for homogeneous networks, but crucially, whenever $\Lambda$ is quasi-homogeneous, which is the case for nearly all realistic networks, then the optimization problems are different, as $\|\theta\|_{\Lambda_{\max}} \neq \|\theta\|$. In the quasi-homogeneous setting, the $\Lambda_{\max}$-seminorm will only depend on a subset of the parameters, and potentially an unexpected subset, such as just the last layer bias parameters for a standard fully-connected network. In section 5 and 6 we will further discuss the implications of this result.

**Intuition.** The heart of the argument proving Theorem 4.1 essentially relies on showing that after all the assumptions are satisfied, then as $t \to \infty$ the $\Lambda$-seminorm diverges $\|\theta\|_\Lambda \to \infty$ and the angle $\omega$ converges $\omega \to 0$. The convergence of $\omega$ implies that the training trajectory converges to a certain characteristic curve and the divergence of $\|\theta\|_\Lambda$ implies that the trajectory diverges along this curve away from the origin. In the homogeneous setting the characteristic curves are rays, implying that as $t \to \infty$ the velocity $\frac{d\theta}{dt}$ aligns in direction to $\theta$. This alignment of the velocity with $\theta = \nabla \frac{1}{2}\|\theta\|^2$ is the key property allowing previous works to derive $\frac{1}{2}\|\theta\|^2$ as the objective function of the implicit optimization problem. However, in the quasi-homogeneous setting, the directions of the characteristic curves are eventually dominated by the $\lambda_{\max}$ parameters, which is what gives rise to the asymmetric objective function $\frac{1}{2}\|\theta\|_{\Lambda_{\max}}^2$ in our work.

**Proof sketch.** We defer most of the technical details of the proof of Theorem 4.1 to App. E, but state the central lemma and the overall logical structure below. As in Lyu & Li (2019), the key mathematical object of our analysis is a *normalized margin*. The *margin*, defined as $q_{\min}(\theta) := \min_i y_i f(x_i; \theta)$, is non-differentiable and unbounded, making it difficult to study. Thus, we define the normalized margin, $\gamma(\theta) := \frac{q_{\min}(\theta)}{\|\theta\|_\Lambda^{\lambda_{\max}^{-1}}}$, and the *smooth normalized margin*, $\tilde{\gamma}(\theta) := \frac{\log((n\mathcal{L})^{-1})}{\|\theta\|_\Lambda^{\lambda_{\max}^{-1}}}$, which is a smooth approximation of $\gamma$. We then prove the following key lemma lower bounding changes in the $\Lambda$-seminorm $\|\theta\|_\Lambda$ and the smooth normalized margin $\tilde{\gamma}$. This lemma holds throughout training, even before separability is achieved, and we believe could be of independent interest to understanding the learning dynamics.

**Lemma 4.1** (Dynamics of $\|\theta\|_\Lambda$ and $\tilde{\gamma}$). *Under assumptions A1, A2, A3, and A4, the dynamics of the $\Lambda$-seminorm and smooth normalized margin are governed by the following inequalities,*

$$
\frac{1}{2}\frac{d}{dt}\|\theta\|_\Lambda^2 \geq \mathcal{L}\log((n\mathcal{L})^{-1}), \qquad \frac{d}{dt}\log(\tilde{\gamma}) \geq \lambda_{\max}^{-1}\frac{d}{dt}\log(\|\theta\|_\Lambda)\tan(\omega)^2,
\tag{2}
$$

*for all $t > 0$ for the first inequality, and for almost every $t > 0$ for the second inequality.*

Notice that once the separability assumption is met, the lower bound on the time-derivative of $\|\theta\|_\Lambda^2$ is strictly positive. This allows us to conclude that the $\Lambda$-seminorm diverges and the loss converges

---

[5]This KKT condition is necessary for the optimality since every feasible point satisfies Mangasarian-Fromovitz constraint qualification (MFCQ) condition (Lemma A.4).

$\mathcal{L} \to 0$ (Lemma E.2). We then seek to prove the directional convergence of the parameters to the tangent vector $\Lambda\theta$ generating the characteristic curves. We first prove that $\tilde{\gamma}$ is upper bounded using the definition of the margin and A7 (Lemma E.3). Combining this upper bound with the monotonicity of $\tilde{\gamma}$ proved in Lemma 4.1, we can conclude by a monotone convergence argument that $\tilde{\gamma}$ will converge. Taken together, the convergence of $\tilde{\gamma}$ and the divergence of $\|\theta\|_\Lambda^2$ implies the angle $\omega \to 0$ on a specific sequence of time (Lemma E.4). Finally, we use the divergence $\|\theta\|_\Lambda \to \infty$ and the convergence $\omega \to 0$ to prove there exists a scaling of the normalized parameters that converges to a first-order KKT point of the optimization problem P in Theorem 4.1.

**Non-uniqueness of** $\Lambda$**.** For a quasi-homogeneous function $f$, the value of $\Lambda$, and the $\lambda_{\max}$ parameter set, is not necessarily unique and therefore one may think Theorem 4.1 looks inconsistent. However, the conditional separability (A7), which is required to apply Theorem 4.1, removes this possibility. See App. B for a discussion on how to determine the highest-rate $\lambda_{\max}$ parameter set.

## 5 QUASI-HOMOGENEOUS MAXIMUM-MARGIN CAN DEGRADE ROBUSTNESS

In section 4 we showed how gradient flow on a quasi-homogeneous model will implicitly minimize the norm of only the highest-rate parameters. To explore the implications that this bias has on function space, we will consider a simple problem where analytic solutions exist. We will analyze the binary classification task of learning a linear classifier $w$ that separates two balls in $\mathbb{R}^d$. Consider a dataset that forms two disjoint dense balls $B(\pm\mu, r)$ with centers at $\pm\mu \in \mathbb{R}^d$ and radii $r \in \mathbb{R}_+$. The label $y_i$ of a data point $x_i$ is determined by which ball it belongs to, such that $y_i = 1$ if $x_i \in B(\mu, r)$ and $y_i = -1$ if $x_i \in B(-\mu, r)$. We assume $\|\mu\| = 1$ and that $r < 1$ to ensure linear separability. We measure the quality of a classifier by its *robustness*, the minimum Euclidean distance between the decision boundary $\{x \in \mathbb{R}^d : \langle w, x \rangle = 0\}$ and the balls $B(\pm\mu, r)$. See Fig. 3 for a depiction of the problem setup.

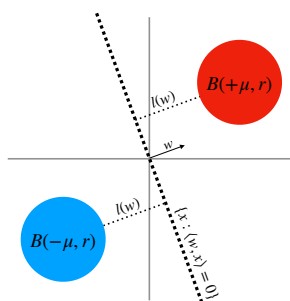

Figure 3: **An illustrative example.** A 2D depiction of the binary classification task of learning a linear classifier $w$ to separate two balls $B(\pm\mu, r)$. The robustness $l(w)$ is measured by the minimum Euclidean distance between the decision boundary and the balls.

We will consider two parameterizations of a linear classifier, one that is homogeneous $f_{\text{hom}}(x; \theta) = \sum_i \theta_i x_i$ and one that is quasi-homogeneous $f_{\text{quasi-hom}}(x; \theta) = \sum_i (\prod_{j=1}^{D_i} \theta_{ij}) x_i$ where $D_i = 1$ for the first $m$-coordinates and $D_i > 1$ for the last $(d - m)$-coordinates. For the quasi-homogeneous model, the parameters associated with the first $m$-coordinates are the $\lambda_{\max}$ parameters. Let $P \in \mathbb{R}^{d \times d}$ be the projection matrix into the subspace spanned by the first $m$-coordinates, $P_\perp = I - P$ be the one into the last $(d-m)$-coordinates, and $\rho_\mu := \|P_\perp \mu\|$ be the norm of $\mu$ projected into this subspace. As long as the radius $r > \rho_\mu$, then the conditional separability assumption of Theorem 4.1 is satisfied[6]. Applying Theorem 4.1, we can conclude that for appropriate initializations[7], $f_{\text{hom}}$ and $f_{\text{quasi-hom}}$ converge to the linear classifiers defined by the following optimization problems respectively,

$$\min_{w \in \mathbb{R}^d} \|w\| \text{ s.t. } y(x) \langle w, x \rangle \geq 1 \quad \forall x \in B(\pm\mu, r), \tag{3}$$

$$\min_{w \in \mathbb{R}^d} \|Pw\| \text{ s.t. } y(x) \langle w, x \rangle \geq 1 \quad \forall x \in B(\pm\mu, r). \tag{4}$$

Each of these two optimization problems is convex and has a unique minimizer, which we can derive exact expressions for by considering the subspace spanned by the vectors $P\mu$ and $P_\perp\mu$.

**Lemma 5.1.** *If separability ($r < 1$) and conditional separability ($r > \rho_\mu$) hold, then Eq. 3 and Eq. 4 have unique minimizers, $w_{hom}$ and $w_{quasi-hom}$ respectively, which satisfy,*

$$w_{hom} \propto \mu, \qquad w_{quasi-hom} \propto \sqrt{\frac{1 - r^{-2}\rho_\mu^2}{1 - \rho_\mu^2}} P\mu + r^{-1}P_\perp\mu, \tag{5}$$

---

[6] For all $w \in \mathbb{R}^d$ that separate the two balls $B(\pm\mu, r)$, $\|Pw\| > 0$.

[7] This problem does not have local minima, but it does have saddle points.

*such that the robustness of these optimal classifiers is*

$$l(w_{hom}) = 1 - r, \qquad l(w_{quasi\text{-}hom}) = \sqrt{1 - r^{-2}\rho_\mu^2} \left( \sqrt{1 - \rho_\mu^2} - \sqrt{r^2 - \rho_\mu^2} \right). \qquad (6)$$

From these expressions it is easy to confirm that $l(w_{\text{quasi-hom}}) \leq l(w_{\text{hom}})$ for all $\rho_\mu < r < 1$. For a fixed $\rho_\mu$, the gap in robustness between the homogeneous and quasi-homogeneous models increases as $r \downarrow \rho_\mu$. These expressions demonstrate that the quasi-homogeneous maximum-margin bias can lead to a solution with vanishing robustness in function space. To confirm this conclusion, we train $f_{\text{hom}}$ and $f_{\text{quasi-hom}}$ with gradient flow and keep track of the classifier $w$ and robustness $l(w)$ for the two models, while sweeping the radius from $\rho_\mu$ to 1. As shown in Fig. 4, we see a sharp drop in the highest-rate parameters ($w_1$) and the robustness of the quasi-homogeneous model as $r \downarrow \rho_\mu (= 0.5)$, while for the homogeneous model, the parameters are stable and the robustness is linear[8] in $r$, as expected from Lemma (5.1).

So far we have restricted our analysis to the setting $r > \rho_\mu$, such that we can be certain the conditional separability assumption is met. But what happens to the performance of the quasi-homogeneous model below this threshold $r \leq \rho_\mu$? As shown in Fig. 4, it appears that the model learns to discard the highest-rate parameters once they are unnecessary and the maximum-margin bias continues on the resulting sub-model. In Fig. 4, when $r \leq 0.5$, the second highest-rate parameters ($w_2$) for the quasi-homogeneous model begins to collapse and the robustness curve repeats another swell, eventually collapsing again when $r = 0.25$. Based on this, we conjecture a stronger version of Theorem 4.1 without the conditional separability assumption. This conjecture is very similar to an informal conjecture discussed in Nacson et al. (2019a) for ensembles of homogeneous models.

**Conjecture 5.1** (Cascading Minimization). *Under assumptions A1 to A6, there exists a $\tilde{\lambda} \in \mathbb{R}_+$ and an $\alpha \in \mathbb{R}$ such that $\psi_\alpha(\lim_{t\to\infty} \hat{\theta}(t))$ is a first-order KKT point of the optimization problem:*

$$\begin{aligned} \textit{minimize} \quad & \frac{1}{2}\|\theta\|^2_{\Lambda I_{\tilde{\lambda}}} \\ \textit{subject to} \quad & y_i f(x_i; \theta) \geq 1 \quad \forall i \in [n] \\ & \theta_l = 0 \quad \forall \lambda_l > \tilde{\lambda}, \end{aligned}$$

*where $I_{\tilde{\lambda}}$ is a diagonal matrix whose entry $(I_{\tilde{\lambda}})_{ii}$ is 1 if $\lambda_i = \tilde{\lambda}$ and 0 otherwise.*

As shown in Fig. 4, we find evidence of a cascading minimization of the first and then second highest-rate parameters as the radius drops below the respective thresholds that make these parameters necessary.

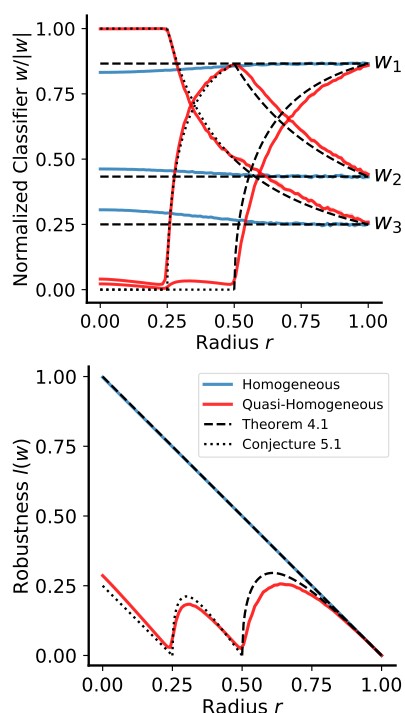

Figure 4: **Asymmetric maximum-margin can collapse robustness.** Tracking the elements of a classifier $w$ and its robustness $l(w)$ for a homogeneous and quasi-homogeneous model, trained by gradient flow on the binary classification problem in $\mathbb{R}^3$ for a sweep of radii $r$. As predicted by Lemma 5.1, for the homogeneous model the classifier $w \propto \mu$ and robustness is linear in $r$, while for the quasi-homogeneous model the highest-rate parameters and the robustness collapses when $r = \rho_\mu = 0.5$. The value of $\mu = [0.87, 0.43, 0.25]$ and $\Lambda = [1, 0.2, 0.1]$. See App. I for experimental details.

## 6   A MECHANISM BEHIND NEURAL COLLAPSE

In this section, we move away from linear models and consider the implications the quasi-homogeneous maximum-margin bias has in the setting of highly-expressive neural networks used

---

[8]If we consider higher-order homogeneous models, such as a deep linear network, then the resulting maximum margin bias would prefer sparse solutions, which could erode the robustness.

in practice. We identify that for sufficiently expressive neural networks with normalization layers, the asymmetric norm minimization drives the network to *Neural Collapse*, an intriguing empirical phenomenon of the last layer parameters and features recently reported by Papyan et al. (2020). In their paper, they demonstrate that the following four properties can be universally observed in the learning trajectories of deep neural networks once the training error converges to zero: (1) The last-hidden-layer feature vector converges to a single point for all the training data with the same class label. (2) The convex hull of the convergent feature vectors forms a *regular $(C-1)$-simplex*[9] centered at the origin, where $C$ is the number of possible class labels. (3) The last-layer weight vector for each class label converges to the corresponding feature vector up to re-scaling. (4) For a new input, the neural network classifies it as the class whose convergent feature vector is closest to the feature vector of the given input.

A considerable amount of effort has been made to theoretically understand this mysterious phenomenon. Han et al. (2021); Poggio & Liao (2019); Mixon et al. (2022); Rangamani & Banburski-Fahey (2022) studied Neural Collapse in the setting of mean-squared loss and Fang et al. (2021); Tirer & Bruna (2022); Weinan & Wojtowytsch (2022); Zhu et al. (2021); Ji et al. (2021) introduced toy models to explain Neural Collapse in the setting of cross-entropy loss. These toy models are optimization problems over the last-hidden-layer feature vectors and the last-layer parameters, but not including parameters in the earlier layers. Many of these works introduced unjustified explicit regularizations or constraints on the feature vectors in their model. A recent work, Ji et al. (2021), showed how gradient dynamics on the space of the last-hidden-layer feature vectors and last-layer weights, without any explicit regularization, would lead to Neural Collapse as a result of the implicit maximum-margin bias. However, the real gradient dynamics of neural networks happen in the space of all parameters of the model, and hence it is not clear how an implicit bias that leads the model to Neural Collapse, can be induced by the *parameter gradient dynamics*.

In this section, we show that the parameter gradient dynamics of any present-day neural networks can universally show Neural Collapse as long as they are sufficiently expressive, apply normalization to the last hidden layer, and are trained with the cross-entropy loss. Our theoretical analysis is based on the regularization by normalization and the quasi-homogeneous maximum-margin bias. Note that in Papyan et al. (2020), all the neural networks showing Neural Collapse are trained with the cross-entropy loss and have normalization.

Specifically, we consider the $C$-class classification model $f_c(x) = w_c^T h(x, \theta') + b_c$ where the last layer weights $w_c \in \mathbb{R}^d$ and bias $b_c$ for $c \in [C]$ and the last-layer feature $h(x, \theta') \in \mathbb{R}^d$. The feature vector $h(x, \theta')$ is obtained with layer normalization[10], and therefore it satisfies

$$\sum_{j=1}^{d} h_j(x_i, \theta') = 0, \quad \sum_{j=1}^{d} h_j^2(x_i, \theta') = 1 \quad \forall i \in [n], \tag{7}$$

where $\{(x_i, y_i)\}_{i \in [n]}$ is the training data. This model is quasi-homogeneous with $\lambda = 1$ for the $w_c$ and $b_c$, and $\lambda = 0$ for parameters in the earlier layers $\theta'$. Thanks to this quasi-homogeneity, our result for multi-class classification tasks (see App. H) reveals that the rescaled parameters converge to a first-order KKT point of the following optimization problem:

$$\min_{(w,b,\theta')} \sum_{c \in [C]} |w_c|^2 + |b|^2 \text{ s.t. } \min_{i \in [n]} \left[ (w_{y_i})^T h(x_i, \theta') + b_{y_i} - \max_{c \neq y_i} \left[ (w_c)^T h(x_i, \theta') + b_c \right] \right] \geq 1. \tag{8}$$

We further make the following assumptions on expressivity and data distribution:

- **A8** (*Sufficient Expressivity*). For any $\{(x_i', h_i')\}_{i \in [n]}$ satisfying $\sum_j (h_i')_j = 0$ and $\sum_j (h_i')_j^2 = 1 \quad \forall i \in [n]$, there exists $\theta'$ satisfying $h(x_i', \theta') = h_i'$ for any $i \in [n]$.
- **A9** (*Existence of All Labels*). For each class $c \in [C]$, there exists at least one data point in $\{(x_i, y_i)\}_{i \in [n]}$ whose label $y_i$ belongs to $c$.

The first assumption is to eliminate the possibility that any parameter configuration $\theta'$ cannot realize Neural Collapse. Under these assumptions, the global minimum satisfies Neural Collapse:

---

[9]A regular $(C-1)$-simplex is the convex hull of $C$ points where the distance between any pair is the same. Papyan et al. (2020) refer to this simplex centered at the origin as a *general simplex Equiangular Tight Frame*.

[10]Here we use layer normalization, but similar theorems would hold for other normalization schemes, such as batch normalization.

**Theorem 6.1** (Neural Collapse, short version). *Under assumptions A8, A9, and $d \geq C$, any global optimum of Eq.8 satisfies the four properties of Neural Collapse.*

Note that we do not exclude the possibility that Eq.8 has saddles or local minima. Therefore, depending on the initialization of the learning dynamics, it may end up with those sub-optimal first-order KKT points, which may not show Neural Collapse.

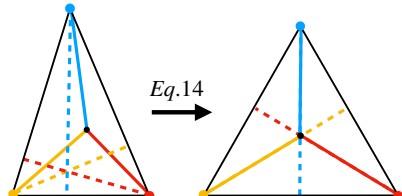

Essentially, the proof of Theorem 6.1 relies on first relaxing Eq. 8 to the optimization problem

$$\min_{(w)} \sum_c |w_c|^2 \text{ s.t. } \min_{c \in [C]} L_c \geq 1, \quad (9)$$

where $L_c$ is the minimum distance from $w_c$ to the $(C-2)$-simplex formed by the convex hull of $\{w_{c'}\}_{c' \in [C]/\{c\}}$. With accordance to our geometric intuition, the minimizer of this optimization problem is a regular $(C-1)$-simplex. See Fig. 5 for a visual depiction of this relaxed optimization problem and App. G for the details of the proof.

Figure 5: **Geometric intuition.** An illustration of Eq. 9. The black circle represents the origin, the solid lines represent the class vectors $w_c$, and the dotted lines represent the distance $L_c$. Intuitively, minimizing the lengths of the solid lines while maintaining a minimum length of the dotted lines will result in a regular simplex centered at the origin.

## 7 CONCLUSION

In this work, we extend and generalize a long line of literature studying the maximum-margin bias of gradient descent to quasi-homogeneous networks. We show that after reaching a point of separability, the gradient flow dynamics are driven by a competition between maximizing the margin in function space and minimizing the $\Lambda_{\max}$-seminorm in parameter space. We demonstrate, with a simple linear example, how this strong favoritism for the highest-rate parameters can degrade the robustness of quasi-homogeneous models and conjecture that this process, when possible, will reduce the model to a sparser parameterization. Additionally, by applying our theorem to sufficiently expressive neural networks with normalization layers, we reveal a universal mechanism behind Neural Collapse. Here we propose some future directions for this work.

**Discretization effect.** In this work, we only considered gradient flow, but generalizing the theoretical results to (stochastic) gradient descent is an important future step. In particular, it is well understood that the discretization effect introduced by a finite learning rate has empirically measurable effects for parameters that are scale-invariant, such as those before normalization layers. While gradient flow would predict the norm of these parameters to be constant through training, gradient descent predicts that they monotonically diverge, as demonstrated by Kunin et al. (2020). Thus, extending our results to the setting of gradient descent could reshape Theorem 4.1.

**Optimality of convergence points.** We are only able to guarantee by Theorem 4.1 that the learning dynamics will converge to a first-order KKT point of the constrained optimization problem, but not whether this point is locally or globally optimal. Better understanding the landscape of this optimization problem and determining when stronger statements can be made is a promising direction for future work. Works such as Chizat & Bach (2020); Ji & Telgarsky (2020); Vardi et al. (2021); Lyu et al. (2021) have made progress in this direction for simple homogeneous networks and could provide a strategy for investigating more complex quasi-homogeneous models.

**Influence of initialization.** A major limitation of analyzing the maximum-margin bias of gradient flow is that the dynamics in this terminal phase of training are slow to converge or only become evident at extremely unpractical training loss levels. Motivated by this limitation, Woodworth et al. (2020) and Moroshko et al. (2020) studied the gradient flow trajectories for diagonal linear networks and showed that there is a transition from a "kernel" regime to a "rich" regime controlled by the scale of the initialization and the final training loss level. Extending this analysis to quasi-homogeneous networks would be a valuable future direction.

**Impact on performance.** An important takeaway from our work is that the maximum-margin bias can actually degrade the performance of a quasi-homogeneous model. The benefit depends on the parameterization of a model and its relationship to the geometry of the data. Better understanding this interaction could be essential for diagnosing performance gaps of modern neural networks and provide a route towards designing robust architectures.

## ACKNOWLEDGMENTS

We thank Kaifeng Lyu, Ben Sorscher, Daniel Soudry, and Hidenori Tanaka for helpful discussions. D.K. thanks the Open Philanthropy AI Fellowship for support. A.Y. thanks the Masason Foundation for support. S.G. thanks the James S. McDonnell and Simons Foundations, NTT Research, and an NSF CAREER Award for support while at Stanford.

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

CONTENTS

## A    MORE DETAILS ON QUASI-HOMOGENEOUS MODELS

$\Lambda$-normalization. Here we provide more details on $\Lambda$-normalization as discussed in section 3. The $\Lambda$-normalized parameters $\hat{\theta}$ are given by

$$\hat{\theta} = \left(e^{-\tau\lambda_1}\theta_1, \ldots, e^{-\tau\lambda_m}\theta_m\right) \quad \text{s.t.} \quad \|\hat{\theta}\|_\Lambda^2 = 1$$

The value of $\tau$ is implicitly defined through the constraint $\|\hat{\theta}\|_\Lambda^2 = 1$. Only in a select number of cases does an explicit expression for $\tau$ exist. For example, in the homogeneous setting when $\Lambda = I$, $\tau = \log(\|\theta\|)$ and $\hat{\theta} = \frac{\theta}{\|\theta\|}$, as would be expected.

**Lemma A.1.** *For all $\theta \in \mathbb{R}^m$ such that $\|\theta\|_\Lambda > 0$, the $\Lambda$-normalized parameters $\hat{\theta}$ are unique.*

*Proof.* Proving uniqueness of $\hat{\theta}$ is equivalent to proving uniqueness of $\tau$. For a given $\theta$ and $\Lambda$, then $\tau = \log(1/\sqrt{z})$ where $z$ is the positive root of the polynomial $\sum_i \lambda_i \theta_i^2 z^{\lambda_i} - 1 = 0$. The coefficients $\lambda_i \theta_i^2 \geq 0$ are non-negative, and because $\|\theta\|_\Lambda > 0$, we know there exists at least one positive coefficient $\lambda_i \theta_i^2 > 0$. Thus, there is exactly one sign change in the coefficients of this polynomial, which by Descartes' rule of signs, implies the polynomial has exactly one positive root, and thus $\tau$ is unique. $\square$

**Locally Lipschitz Quasi-homogeneous models.** To apply our analysis and Theorem 4.1 to many deep neural network settings including those with non-smooth ReLU activations, we here consider quasi-homogeneous functions with local Lipschitz property. For such functions $f(\theta) : \mathbb{R}^d \to \mathbb{R}$, Clarke's subdifferential $\partial_\theta^\circ$ is defined as follows Clarke et al. (2008).

**Definition A.1** (Clarke's subdifferential).

$$\partial_\theta^\circ f(\theta) := \text{conv} \left\{ \lim_{k\to\infty} \nabla_\theta f(\theta_k) : \lim_{k\to\infty} \theta_k = \theta, f \text{ is differentiable at } \theta_k \right\}. \quad (10)$$

Similar to Theorem B.2 in Lyu & Li (2019), we can show that $\partial_\theta^\circ f(\theta)$ satisfies a scaling property and a version of Euler's theorem.

**Lemma A.2.** *Let $f(\theta) : \mathbb{R}^d \to \mathbb{R}$ be locally Lipschitz and $\Lambda$-quasi-homogeneous. $\partial_\theta^\circ f$ satisfies the following scaling property:*

$$\partial_\theta^\circ f(\psi_\alpha(\theta)) = \left\{ e^{\alpha(I-\Lambda)} h : h \in \partial_\theta^\circ f(\theta) \right\} \quad (11)$$

*for any $\alpha > 0$ and $\theta \in \mathbb{R}^d$.*

*Proof.* For any sequence $\{\theta_k\}_{k\in\mathbb{N}}$ on which $f$ is differentiable and converging to $\theta$, $\{\psi_\alpha(\theta_k)\}_{k\in\mathbb{N}}$ converges to $\psi_\alpha(\theta)$ and $f$ is differentiable on this new sequence, whose derivative is given by

$$\nabla_\theta f(\psi_\alpha(\theta_k)) = e^\alpha \left.\frac{\partial(e^{-\alpha}f(\theta))}{\partial\theta}\right|_{\psi_\alpha(\theta_k)} = e^\alpha \left.\frac{\partial f(e^{-\alpha\Lambda}\theta)}{\partial\theta}\right|_{\psi_\alpha(\theta_k)} = e^{\alpha(I-\Lambda)} \left.\frac{\partial f(e^{-\alpha\Lambda}\theta)}{\partial e^{-\alpha\Lambda}\theta}\right|_{\psi_\alpha(\theta_k)},$$

where the last expression is equivalent to $e^{\alpha(I-\Lambda)}\nabla_\theta f(\theta_k)$. Conversely, for any sequence $\{\psi_\alpha(\theta_k)\}_{k\in\mathbb{N}}$ on which $f$ is differentiable and converging to $\psi_\alpha(\theta)$, $\{\theta_k\}_{k\in\mathbb{N}}$ converges to $\theta$ and $f$ is differentiable on it as well, with the above scaling property. Hence,

$$\left\{ \lim_{k\to\infty} \nabla_\theta f(\theta_k) : \lim_{k\to\infty} \theta_k = \psi_\alpha(\theta), f \text{ is differentiable at } \theta_k \right\}$$
$$= \left\{ e^{\alpha(I-\Lambda)} \lim_{k\to\infty} \nabla_\theta f(\theta_k) : \lim_{k\to\infty} \theta_k = \theta, f \text{ is differentiable at } \theta_k \right\}.$$

Thus taking the convex hulls of both sets, and by the commutativity between conv and the linear operation $e^{\alpha(I-\Lambda)}$, we conclude that Eq.11 holds. $\square$

By further assuming that $f(\theta)$ admits a chain rule (See Davis et al. (2020) or Lyu & Li (2019) for its definition), we can show that $f$ satisfies a version of Euler's theorem, similar to the case of homogeneous functions.

**Lemma A.3.** *If $f(\theta) : \mathbb{R}^d \to \mathbb{R}$ is locally Lipschitz admitting a chain rule and $\Lambda$-quasi-homogeneous, then it satisfies a version of Euler's theorem, i.e., for all $\theta \in \mathbb{R}^d$,*

$$\langle h, \Lambda\theta \rangle = f(\theta) \text{ for any } h \in \partial_\theta^\circ f(\theta). \tag{12}$$

*Proof.* Since $f$ admits a chain rule, there exists $\alpha > 0$ such that $f(e^{\alpha\Lambda}\theta)$ is differentiable with respect to $\alpha$ and

$$\frac{d}{d\alpha} f(e^{\alpha\Lambda}\theta) = \left\langle g, \frac{de^{\alpha\Lambda}\theta}{d\alpha} \right\rangle = \left\langle g, e^{\alpha\Lambda}\Lambda\theta \right\rangle,$$

for any $g \in \partial_\theta^\circ f(e^{\alpha\Lambda}\theta)$. Therefore

$$f(\theta) = e^{-\alpha} \frac{d}{d\alpha} \left( e^\alpha f(\theta) \right) = e^{-\alpha} \frac{d}{d\alpha} f(e^{\alpha\Lambda}\theta) = \left\langle e^{-\alpha(I-\Lambda)} g, \Lambda\theta \right\rangle.$$

By Lemma A.2, for any $h \in \partial_\theta^\circ f(\theta)$, we can find $g \in \partial_\theta^\circ f(e^{\alpha\Lambda}\theta)$, such that

$$\langle h, \Lambda\theta \rangle = \left\langle e^{-\alpha(I-\Lambda)} g, \Lambda\theta \right\rangle = f(\theta).$$

$\square$

The properties above immediately implies that any feasible point of optimization problem (P) satisfies Mangasarian-Fromovitz constraint qualification (MFCQ) condition, which implies the first-order KKT condition is necessary for the optimality.

**Lemma A.4.** *Any feasible point $\theta$ of optimization problem (P) satisfies Mangasarian-Fromovitz constraint qualification (MFCQ) condition, i.e., there exists $v \in \mathbb{R}^d$ such that for all $i \in [n]$ with $y_i f(x_i, \theta) = 1$,*

$$\langle v, h \rangle > 0 \text{ for any } h \in \partial_\theta^\circ (y_i f(x_i, \theta) - 1).$$

*Proof.* Notice that for any $h \in \partial_\theta^\circ (y_i f(x_i, \theta) - 1)$, there exists $h' \in \partial_\theta^\circ f(x_i, \theta)$ such that $\langle \Lambda\theta, h \rangle = y_i \langle \Lambda\theta, h' \rangle$. Hence, choosing $v = \Lambda\theta$, by Lemma A.3,

$$\langle \Lambda\theta, h \rangle = y_i \langle \Lambda\theta, h' \rangle = y_i f(x_i, \theta) = 1 > 0.$$

$\square$

**Ensembles of homogeneous models.** In Nacson et al. (2019a), they considered the maximum-margin bias of gradient descent for non-homogeneous models that can be expressed as finite sums of positive-homogeneous models of different orders. In particular, for some $K \in \mathbb{N}$, they consider functions $f(x; \theta)$ that can be expressed as

$$f(x; \theta) = \sum_{k=1}^K f^{(k)}(x; \theta_k), \tag{13}$$

where $\theta = [\theta_1, \ldots, \theta_K]$ and $f^{(k)}(x; \theta_k)$ is $\alpha_k$-positive homogeneous such that $0 < \alpha_1 < \cdots < \alpha_K$. While this class of models is not homogeneous because of the varying orders of the sub-models, it is quasi-homogeneous. If we choose $\Lambda$ such that for all parameters in $\theta_k$ the value of $\lambda = \alpha_k^{-1}$, then $f(x; \theta)$ is $\Lambda$-Quasi-Homogeneous. Therefore, the theoretical results discussed in this work should align with the results discussed in Nacson et al. (2019b) for the setting of ensembles of positive-homogeneous models. Indeed Theorem 4.1 and Conjecture 5.1 agree with analysis stated in their work that "an ensemble on neural networks will aim to discard the shallowest network in the ensemble", which is the sub-model with the highest-rate parameters.

While all ensembles of positive-homogeneous models are quasi-homogeneous, not all quasi-homogeneous models are ensembles. Here we provide a short list of quasi-homogeneous models that cannot be written in the form of Eq. 13.

*Deep fully connected network with biases.* Consider again the two-layer fully connected network with biases discussed in section 3,

$$f(x; \theta) = w^2 \sigma \left( \sum_i w_i^1 x_i + b^1 \right) + b^2.$$

If we arrange terms such that $f^{(1)}(x; b^2) = b^2$ and $f^{(2)}(x; w^1, b^1, w^2) = w^2\sigma\left(\sum_i w_i^1 x_i + b^1\right)$, then we can express $f(x; \theta) = f^{(1)}(x; b^2) + f^{(2)}(x; w^1, b^1, w^2)$, which is an ensemble of two positive-homogeneous models with $\alpha_1 = 1$ and $\alpha_2 = 2$. However, notice that if we consider a third layer with parameters $w^3$ and $b^3$, then this decoupling of the network is not possible unless some of the sub-models share parameters, preventing us from expressing $f(x; \theta)$ in the form of Eq. 13. All fully connected networks with biases, and a depth greater than two, are quasi-homogeneous models, but not an ensemble of positive-homogeneous models.

*Networks with degenerate* $\Lambda$. As presented earlier, for quasi-homogeneous networks with residual connections or normalization layers, we can choose $\Lambda$ to have zero values. Thus, even if these networks could be decoupled into a sum of sub-models that don't share parameters, the sub-models associated with the zero $\lambda$ parameters would not be positive-homogeneous.

In summary, the results presented in this work coincide with the results presented in Nacson et al. (2019a) for ensembles of positive-homogeneous models, but also apply to a far more general class of non-homogeneous models.

# B    THE CONSISTENCY OF THEOREM 4.1 AND THE PROPER CHOICE OF $\Lambda$

As briefly discussed in section 4, for a quasi-homogeneous function $f$, the value of $\Lambda$, and the $\lambda_{\max}$ parameter set, is not necessarily unique and therefore one may think Theorem 4.1 looks inconsistent. However, the conditional separability (A7), which is required to apply Theorem 4.1, removes this possibility. Here we provide some insightful examples and then provide a complete proof.

**Examples of quasi-homogeneous models with non-unique $\Lambda$.** We here clarify through examples how our theorem can be consistent with cases where the model $f(\theta)$ is quasi-homogeneous with multiple choice of $\Lambda$ due to additional symmetry.

*A linear model with two parameters.* We consider the following model,

$$f(x; \theta_1, \theta_2) = \theta_1 \theta_2^2 x.$$

This is quasi-homogeneous with $(\lambda_1, \lambda_2) = (1-2\xi, \xi)$ for any $\xi \in [0, 1/2]$. There are three possible sets of parameters with largest $\lambda$ value:

- If $\xi > 1/3$, $\theta_1$ has the largest $\lambda$ value.
- If $\xi < 1/3$, $\theta_2$ has the largest $\lambda$ value.
- If $\xi = 1/3$, $\theta_1$ and $\theta_2$ have the same $\lambda$ value.

If one naively applies the theorem to these cases, they might think that the learning process converges to a separable solution minimizing $\theta_1^2$ for the first case, $\theta_2^2$ for the second case, and $\theta_1^2 + \theta_2^2$ for the latter case, which is inconsistent. However, the first two cases do not satisfy the conditional separability assumption. This is because we can make $|\theta_1|$ or $|\theta_2|$ as small as possible while fixing the function itself. Therefore the correct choice of $\lambda$ should be $(\lambda_1, \lambda_2) = (1/3, 1/3)$.

*Two-layer quadratic activation with biases.* For the sake of simplicity, we assume that all the layer widths are one, i.e., the model is given by four scalar parameters as follows:

$$f(x; \theta) = \theta_3 \left(\theta_1 x + \theta_2\right)^2 + \theta_4.$$

We can easily generalized our argument to wider networks. This model is quasi-homogeneous with the following choices of $\lambda$:

$$(\lambda_1, \lambda_2, \lambda_3, \lambda_4) = (\xi, \xi, 1 - 2\xi, 1) \text{ for any } \xi \in [0, 1/2].$$

Again, there are three possibilities.

- If $\xi = 0$, $\theta_3, \theta_4$ have the largest $\lambda$ value.
- If $\xi \in (0, 1/2)$, $\theta_4$ has the largest $\lambda$ value.
- If $\xi = 1/2$, $\theta_1, \theta_2, \theta_4$ have the largest $\lambda$ value.

All of the cases can satisfy the conditional separability condition. For the first case, our theorem tells that the gradient dynamics minimizes $\theta_3^2 + \theta_4^2$. However, by making $\theta_1$ and $\theta_2$ large, we can make $\theta_3$ arbitrary small without changing the output, and hence, it is equivalent to minimizing $\theta_4^2$ alone. This argument also holds for the third case. Thus, for all three cases $\theta_4^2$ is the objective function for the minimization.

*A neural network with normalization.* We consider the following model,

$$f(x; \theta) = \sum_{c \in [C]} w_c^T F_{\text{norm}}(h(\theta', x)) + b,$$

where $w_c \in \mathbb{R}^d$, $b \in \mathbb{R}^C$ are the weight and bias on the last layer, $\theta'$ is the set of parameters in the earlier layers, and $h(\theta', x) \in \mathbb{R}^d$ is the feature vector on the last hidden layer, which we assume is homogeneous[11], i.e., $e^\alpha h(\theta'; x) = h(e^{\alpha \lambda'} \theta'; x)$ for any $\alpha \in \mathbb{R}$ with a certain $\lambda' > 0$. $F_{\text{norm}}(\cdot)$ is a normalizer of the feature vector $h(\theta'; x)$ so that the normalized feature vector $F_{\text{norm}}(h(\theta'; x))$ is invariant under scaling transformation of $\theta'$, i.e., $h(\theta'; x) = h(e^{\alpha \lambda'} \theta'; x)$ for any $\alpha \in \mathbb{R}$. In this setting, possible choices of $\lambda$ values are 1 for the last layer parameters and $\xi$ for parameter $\theta_i'$ where $\xi$ is an arbitrary non-negative number. Thus there are at least following three possible sets of parameters with largest $\lambda$.

---

[11]Our discussion here works with quasi-homogeneity, but we assume homogeneity here for simplicity.

- If $\xi > 1$, the parameters $\theta'$ in the earlier layers have the largest $\lambda$.
- If $\xi = 1$, all the parameter have the same value of $\lambda$.
- If $\xi < 1$, the last-layer weights and biases have the largest $\lambda$.

In the first case, by the scale invariance of $h(\theta'; x)$, we can make $\|\theta'\|$ as small as possible while not changing $f(\theta, x)$, which implies that it does not satisfy the conditional separability condition. On the other hand, in the second case, it satisfies the conditional separability, since there need to be non-zero last-layer weights or bias to correctly classify data points. By applying the theorem, we can conclude that the learning process converges to a minimizer of $\sum_c \|w_c\|^2 + \|b\|^2 + \|\theta'\|^2$. However, we can minimize $\|\theta'\|$ as much as we want, while fixing $f(\theta; x)$, and hence it is equivalent to minimizing $\sum_c \|w_c\|^2 + \|b\|^2$. In the third case, we can apply the theorem as well, which means that the learning process converges to a minimizer of sum of $\sum_c \|w_c\|^2 + \|b\|^2$.

In summary, while the choice of $\Lambda$ is not necessarily unique because of intrinsic symmetries in the parameterization of the model, the set of highest-rate parameters is well defined by the constraints imposed by the conditional separability assumption. This makes Theorem 4.1 well defined.

**A complete proof of uniqueness of the resulting optimization problem** As we discussed above going through three examples, we can identify which $\lambda$ we should choose, or which $\lambda_{\max}$ parameters we should choose, solely by analyzing the model, independent of the data set. In this section, we generalize the previous discussions on examples and prove that the resulting first-order KKT condition derived from Theorem 4.1 is unique, even if the model itself is quasi-homogeneous with multiple choices of scaling parameters $\Lambda$. Note that the following argument is independent of the data and properties derived here in this section is solely the properties of the architecture of the model itself.

In the following argument, to simplify our proof, we assume $f(s; \theta)$ is differentiable with respect to $\theta$, in addition to its continuity. Let $S \subset \mathbb{R}^d$ be the set of $\{\lambda_i\}_{i \in [d]}$ with which the model satisfies the quasi-homogeneity, i.e.,

$$S := \{\lambda \in \mathbb{R}_{\geq 0}^d : f(x; \theta) \text{ is } \lambda\text{-quasihomogeneous}\}.$$

**Definition B.1.** $\lambda \in S$ is called proper if there exists a separable data set $\{(x_i, y_i)\}_{i \in [n]}$ with which the model satisfies A7 and $f(x; \theta)$ is bounded in $\{\theta \in \mathbb{R}^d : \|\theta\|_{\Lambda_{\max}} = 1\}$.

**Definition B.2.** Let $\lambda^1, \lambda^2 \in S$ be proper. We say they are equivalent in terms of first-order KKT conditions, if for any data set $\{(x_i, y_i)\}_{i \in [n]}$, the sets of first-order KKT points for the following two optimization problems

$$\begin{aligned} \textit{minimize} \quad & \frac{1}{2}\|\theta\|_{\Lambda_{\max}^k}^2 \\ \textit{subject to} \quad & y_i f(x_i; \theta) \geq 1 \quad \forall i \in [n] \end{aligned} \tag{14}$$

are equivalent with $k = 1, 2$.

We are going to prove the following theorem in this section.

**Theorem B.1.** All proper points in $S$ are equivalent in terms of their first-order KKT conditions.

Before going to the proof of this theorem, we prove several lemmas for preparation.

**Lemma B.1.** $S$ is convex.

*Proof.* Let $\lambda^1, \lambda^2 \in S$. It suffices to show that for any $\alpha \in [0, 1]$, $\alpha\lambda^1 + (1 - \alpha)\lambda^2 \in S$. By the quasi-homogeneity of $f(x; \theta)$ with respect to $\lambda^1$ and $\lambda^2$, for any $\beta \in \mathbb{R}$,

$$f(x; e^{\beta(\alpha\lambda^1 + (1-\alpha)\lambda^2)}\theta) = e^{\beta - \alpha\beta}f(x; e^{\alpha\beta\lambda^1}\theta) = e^{\beta}f(x; \theta).$$

Hence, $f(x; \theta)$ is $\alpha\lambda^1 + (1 - \alpha)\lambda^2$-quasihomogeneous, i.e., $\alpha\lambda^1 + (1 - \alpha)\lambda^2 \in S$. $\square$

Consider a non-empty line segment in $S$

$$L := \{y \in \mathbb{R}_{\geq 0}^d : y = \zeta t + \lambda^0 \text{ with some } t \in \mathbb{R}\} \subset S, \tag{15}$$

where $\zeta, \lambda^0 \in \mathbb{R}^d$. This set is connected since $S$ is convex. Moreover, it has at least a single end point, because $L \in \mathbb{R}_{\geq 0}^d$. Hence, without loss of generality, we can assume that $\lambda^0$ is the end point and $t$ takes non-negative values. We define a set of indexes $I \subset [d]$ by

$$I := \left\{ i \in [d] : \lambda_i^0 = 0, \exists \lambda \in L, \lambda_i > 0 \right\}.$$

Furthermore, for $\lambda \in S$, we define $M_\lambda$ by

$$M_\lambda := \{ i \in [d] : \lambda_i = \max_{j \in [d]} \lambda_j \}.$$

**Lemma B.2.** *$I$ is non-empty, if $L$ contains at least two different points.*

*Proof.* Suppose $I$ is empty. In the following, we will show that there exists $t < 0$ such that $y(t) = \zeta t + \lambda^0 \in S$. Since this contradicts with the fact that $\lambda^0$ is an end point of $L$, we conclude $I$ is non-empty.

Since $L$ contains at least two different points, there exists $t^1 > 0$ such that $\lambda^1 := y(t^1) \in L$. By quasi-homogeneity of the model with respect to $\lambda^0$ and $\lambda^1$, for any $\alpha, t \in \mathbb{R}$,

$$f(x; e^{\alpha y(t)}\theta) = f(x; e^{\alpha(\frac{t}{t^1}(\lambda^1 - \lambda^0) + \lambda^0)}\theta) = f(x; e^{\alpha \frac{t}{t^1}\lambda^1} e^{\alpha \frac{t^1-t}{t^1}\lambda^0}\theta)$$

$$= e^{\alpha \frac{t}{t^1}} e^{\alpha \frac{t^1-t}{t^1}} f(x; \theta) = e^\alpha f(x; \theta). \tag{16}$$

Since we assume $I$ is empty, for any $i \in [d]$ such that $\zeta_i \neq 0$, $\lambda_i^0 > 0$. By the continuity of $y(t)$ with respect to $t$, this means that there exists an open neighborhood of $t = 0$ where $y_i(t) > 0$ for any $i \in [d]$ such that $\zeta_i \neq 0$. For the other indexes, i.e. $i \in [d]$ such that $\zeta_i = 0$, clearly $y_i(t) \geq 0$ in the open neighborhood. Therefore, in the neighborhood, $y_i(t) \geq 0$ for all $i \in [d]$. In particular, there exists $t < 0$ such that $y(t)_i \geq 0$ for any $i \in [d]$. Combining this fact with Eq.16, we conclude that there exists $t < 0$ such that $y(t) \in S$. By contradiction, $I$ is non-empty. $\square$

**Lemma B.3.** *For any proper element $\lambda^* \in L/\{\lambda^0\}$,*

$$M_{\lambda^*} \cap I \neq \emptyset, \tag{17}$$

*and hence*

$$\max_{i \in [d]} \lambda_i^* = t^* \max_{i \in I} \zeta_i. \tag{18}$$

*Proof.* Since $\lambda^*$ is proper, there exists a data set with which the model satisfies the conditional separability, i.e., there exists $\kappa > 0$ such that all the parameter values $\{\theta_i\}_{i \in [d]}$ which separate the data satisfies $\|\theta\|_{\Lambda_{\max}^*} > \kappa$. Let $\{\theta_i^*\}_{i \in [d]}$ be a parameter values separating the data. By the continuity of the model, without loss of generality, we can assume that $\theta_i^* \neq 0$ for any $i \in [d]$.

In the following, we show $M_{\lambda^*} \cap I \neq \emptyset$ by contradiction. Suppose $M_{\lambda^*} \cap I = \emptyset$. We derive the contradiction by showing that there exists a parameter value $\theta'$ which correctly separates the data, but breaks the conditional separability condition with $\lambda^*$. This clearly contradicts with the fact that $\lambda^*$ is proper.

We consider the following transformation

$$\theta^* \to e^{-\alpha y(0)}\theta^*,$$

with some $\alpha > 0$. This transformation does not change $\{\theta_i^*\}_{i \in I}$, but other parameters are scaled down to $\theta_i^* \to e^{-\alpha \lambda_i^0}\theta_i^*$. Then $\sum_{i \in [d]/I} \lambda_i^* (e^{-\alpha \lambda_i^0}\theta_i^*)^2$ can be arbitrarily small by taking $\alpha \to \infty$. (Notice that here we exploited the fact that if $\lambda_i^0 = 0$, $\lambda_i^* = 0$ for any $i \in [d]/I$.) On the other hand, since $\lambda_i^* > 0$ and $|\theta_i^*| > 0$ for any $i \in I$,

$$\|e^{-\alpha y(0)}\theta^*\|_{\Lambda^*} = \sum_{i \in [d]} \lambda_i^* (e^{-\alpha y(0)}\theta^*)^2 \geq \sum_{i \in I} \lambda_i^* (e^{-\alpha y(0)}\theta^*)^2$$

is lower bounded by a positive constant. (Here we exploit the fact that $I$ is non-empty by Lemma B.2.) Hence by further transforming the parameter

$$e^{-\alpha y(0)}\theta^* \to \theta' := e^{-\beta \Lambda^*} e^{-\alpha y(0)}\theta^*,$$

with a proper choice of $\beta > 0$, we can normalize the $\Lambda^*$-seminorm $\|\theta'\|_{\Lambda^*} = 1$ while keeping $\sum_{i \in [d]/I} \lambda_i^*(\theta_i')^2$ arbitrarily small. In particular, it is smaller than $\kappa$ with large enough $\alpha > 0$. By assumption, we know $[d]/I \supset \mathrm{argmax}_{i \in [d]} \lambda_i^*$, and hence,

$$\|\theta'\|_{\Lambda^*_{\max}}^2 \le \sum_{i \in [d]/I} \lambda_i^*(\theta_i')^2 < \kappa.$$

By the quasi-homogeneity of the model, the model classifies the data set correctly even with this transformed parameter $\theta'$. However, this means that it breaks the conditional separability condition, which contradicts with the fact that $\lambda^*$ is proper. Therefore, $M_{\lambda^*} \cap I \ne \emptyset$.

Lastly $\max_{i \in [d]} \lambda_i^* = t^* \max_{i \in I} \zeta_i$ can be derived as follows:

$$\max_{i \in [d]} \lambda_i^* = \max_{i \in I} \lambda_i^* = t^* \max_{i \in I} \zeta_i.$$

$\square$

**Lemma B.4.** *For any proper element $\lambda^* \in L/\{\lambda^0\}$,*

$$M_{\lambda^*} \cap ([d]/I) \ne \emptyset.$$

*Proof.* Suppose $M_{\lambda^*} \cap ([d]/I) = \emptyset$, i.e., $M_{\lambda^*} \subset I$. Let $\{\theta_i^*\}_{i \in [d]}$ be a parameter values separating the data. Notice that there exists $i \in [d]/I$ such that $\lambda_i^0 \ne 0$ and $\theta_i^* \ne 0$. Otherwise,

$$ef(x; \theta^*) = f(x; e^{y(t^*)}\theta^*) = f(x; e^{2y(t^*/2)}\theta^*) = e^2 f(x; \theta^*),$$

which is a contradiction. We denote such an index by $j$. We consider the following transformation

$$\theta^* \to \theta' := e^{-\beta y(t^*)} e^{\alpha y(0)} \theta^*,$$

for some $\alpha > 0$. $\beta > 0$ here is chosen to satisfy $\|\theta'\|_{\Lambda^*_{\max}} = 1$. By the quasi-homogeneity of the model, the model still correctly classifies the data with this transformed parameters. By taking $\alpha$ arbitrarily large, $(e^{\alpha y(0)}\theta^*)_j = e^{\alpha \lambda_j^0}\theta_j^*$ becomes arbitrarily large, and thus $\beta$ needs to be arbitrarily large to renormalize the $\Lambda^*$-seminorm. Therefore for any $i \in I$, $\theta_i' = e^{-\beta \zeta_i}\theta_i^*$ can be arbitrarily small. Therefore, we can find a large enough $\alpha$ such that

$$\|\theta'\|_{\Lambda^*_{\max}}^2 = \sum_{i \in \mathrm{argmin}\, \lambda_i^*} \lambda_i^*(\theta_i^*)^2 \le \sum_{i \in I} \lambda_i^*(\theta_i^*)^2 < \kappa$$

This contradicts with the fact that $\lambda^*$ is proper. Therefore, $M_{\lambda^*} \cap ([d]/I) \ne \emptyset$. $\square$

**Lemma B.5.** *If there exists a proper $\lambda \in \mathrm{Int}\, S$, it is unique, where $\mathrm{Int}\, S$ is the interior of $S$.*

*Proof.* Suppose there exists two different proper point $\lambda^1, \lambda^2 \in \mathrm{Int}\, S$. In the following argument, we will show that $\lambda^1 = \lambda^2$, which is a contradiction. We consider a line including the two points $\lambda^1$ and $\lambda_2$. Without loss of generality it can be represented as

$$L := \{y \in \mathbb{R}_{\ge 0}^d : y = \zeta t + \lambda^0 \text{ with some } t \ge 0\}, \tag{19}$$

where $\zeta = \lambda^2 - \lambda^1$ and $\lambda^0$ is an end point of this line. Let $\lambda^* = y(t^*)$ be a proper point in the interior of the line. $\lambda^*$ can be either $\lambda^1$ or $\lambda^2$. In the following, we will derive an explicit formula which uniquely determines $t^*$, implying $\lambda^1 = \lambda^2$. By applying Lemma B.3, we obtain

$$\max_{i \in [d]} \lambda_i^* = t^* \max_{i \in I} \zeta_i.$$

Suppose the line has the other end point $t = t_{\max} > 0$. We can apply Lemma B.3 again with this other end point, and we can derive the corresponding equality

$$\lambda_{\max}^* = (t_{\max} - t^*) \max_{i \in J} \zeta_i,$$

where $J \subset [d]$ is given by

$$J := \{j \in [d] : y(t_{\max})_j = 0, \exists \lambda \in L, \lambda_j > 0\}.$$

By comparing these two equality, we obtain

$$t^* = \frac{\max_{i \in J} \zeta_i}{\max_{i \in J} \zeta_i + \max_{i \in I} \zeta_i} t_{\max}.$$

This uniquely determines $t^*$ and hence $\lambda^1 = \lambda^2$.

Next, we consider the other case where $L$ does not have end point other than $\lambda^0$. Notice that

$$\max_{i \in I} \zeta_i > \max_{i \in [d]/I} \zeta_i. \tag{20}$$

This is due to the following reason: Suppose there exists $j \in [d]/I$ such that $\zeta_j \geq \max_{i \in I} \zeta_i$. Since $\zeta_j > 0$, $j \notin I$ implies that $\lambda_j^0 > 0$. Hence, $y(t^*)_j = t^* \zeta_j + \lambda_j^0 > \lambda_{\max}^*$, which is clearly a contradiction.

The inequality Eq.20 means that for any $j \in [d]/I$,

$$\{t \geq 0 : \max_{i \in I} y(t)_i > y(t)_j\} = [t_j', \infty)$$

with some $t_j' \geq 0$. Hence

$$t^* \in \{t \geq 0 : \max_{i \in I} y(t)_i = \max_{i \in [d]} y(t)_i\} = \cap_{j \in [d]/J} \{t \geq 0 : y(t)_J > y(t)_j\} = [t', \infty),$$

where $t' = \max_{j \in [d]/I} t_j'$.

In the region $(t', \infty)$, $\operatorname{argmax}_{i \in [d]} y(t)_i \subset I$, and hence by Lemma B.4, $t^* \notin (t', \infty)$. Hence $t^* = t'$. The uniqueness of $t'$ implies that $\lambda^1 = \lambda^2$.

The argument above shows that in any case $\lambda^1 = \lambda^2$, which contradicts with the assumption that they are different. Therefore, the claim follows. $\qquad\square$

**Lemma B.6.** *For any proper element $\lambda^* \in S$ and any $\lambda \in S$,*

$$\min_{i \in M_{\lambda^*}} (\lambda_i - \lambda_i^*) \leq 0, \quad \max_{i \in M_{\lambda^*}} (\lambda_i - \lambda_i^*) \geq 0.$$

*Proof.* we show this by contradiction. Suppose there exist $\lambda, \lambda^* \in S$ such that $\min_{i \in M_{\lambda^*}} (\lambda_i - \lambda_i^*) > 0$ or $\max_{i \in M_{\lambda^*}} (\lambda_i - \lambda_i^*) < 0$. Let $\theta^*$ be parameter values with which the model can separate a data set. We consider a transformation $\theta \to e^{\alpha y(t)} \theta$ with

$$y(t) = t(\lambda^* - \lambda) + \lambda$$

with some $\alpha, t \in \mathbb{R}$. By quasihomogeneity of the model with respect to $\lambda$ and $\lambda^*$, we know that $f(x; e^{\alpha y(t)} \theta) = e^\alpha f(x; \theta)$. By assumption, there exists $t \in \mathbb{R}$ such that $y_i(t) \leq 0$ for any $i \in M_{\lambda^*}$. This implies that $\{f(x; e^{\alpha y(t)} \theta) : \|e^{\alpha y(t)} \theta\|_{\Lambda_{\max}^*} \leq \|\theta\|_{\Lambda_{\max}^*}, \alpha, t \in \mathbb{R}\}$ is unbounded, which contradicts with our assumption. Hence, the claim follows.

$\qquad\square$

**Lemma B.7.** *Let $\lambda^1, \lambda^2 \in S$ be proper. If the following two conditions are met, $\lambda^1$ and $\lambda^2$ are equivalent in terms of the first-order KKT condition.*

$$\begin{cases} \text{Either } \min_{i \in M_{\lambda^k}} (\lambda_i^1 - \lambda_i^2) = 0 \text{ or } \max_{i \in M_{\lambda^k}} (\lambda_i^1 - \lambda_i^2) = 0 \text{ holds for both } k = 1, 2 \\ \{i \in M_{\lambda^1} : \lambda_i^1 = \lambda_i^2\} = \{i \in M_{\lambda^2} : \lambda_i^1 = \lambda_i^2\} \end{cases}$$

*Proof.* Let $L = \{i \in M_{\lambda^1} : \lambda_i^1 = \lambda_i^2\} (= \{i \in M_{\lambda^2} : \lambda_i^1 = \lambda_i^2\})$. It suffices to show that set of the first-order KKT points of the following problem

$$\begin{aligned} \text{minimize} \quad & \frac{1}{2} \|\theta\|_{\Lambda_{\max}^k}^2 \\ \text{subject to} \quad & y_i f(x_i; \theta) \geq 1 \quad \forall i \in [n] \end{aligned} \tag{21}$$

is equivalent to the set of the first-order KKT points of the following problem

$$
\begin{aligned}
\text{minimize} \quad & \frac{1}{2} \sum_{i \in L} \lambda_{\max}^k \theta_i^2 \\
\text{subject to} \quad & y_i f(x_i; \theta) \geq 1 \quad \forall i \in [n] \\
& \theta_i = 0 \quad \forall i \in (M_{\lambda^1} \cup M_{\lambda^2})/L.
\end{aligned}
\tag{22}
$$

for any data set $\{(x_i, y_i)\}_{i \in [n]}$ for $k = 1, 2$. Since the second optimization problem is clearly a restriction of the first optimization problem with some additional constraints, it suffices to show that any first-order KKT point of the first problem satisfies the constraint $\theta_i = 0$ for any $i \in (M_{\lambda^1} \cup M_{\lambda^2})/L$.

To prove this statement, first we show that for $k = 1, 2$, and any $\theta \in \mathbb{R}^d$, a one-parameter family of parameter values $\{\Theta(\alpha) := e^{-\alpha(\Lambda^1 - \Lambda^2)} \theta : \alpha \in \mathbb{R}\}$ satisfies

- $\frac{d}{d\alpha} f(x; \Theta(\alpha)) = 0$

- If $\frac{d}{d\alpha}\big|_{\alpha=0} \|\Theta(\alpha)\|_{\Lambda_{\max}^k} = 0$, $\theta_i = 0$ for any $i \in M_{\lambda^k}/L$.

The first point can be easily seen by the quasi-homogeneity of the model with respect to $\lambda^1$ and $\lambda^2$. Indeed, for any $\alpha \in \mathbb{R}$,

$$
f(x; \Theta(\alpha)) = f(x; e^{-\alpha\Lambda^1} e^{\alpha\Lambda^2} \theta) = e^{-\alpha} f(x; e^{\alpha\Lambda^2} \theta) = f(x; \theta).
$$

Regarding the second point, first notice that for any $i \in L$, $\Theta_i(\alpha) = \theta_i$ for any $\alpha \in \mathbb{R}^d$ and hence

$$
\frac{d}{d\alpha}\bigg|_{\alpha=0} \|\Theta(\alpha)\|_{\Lambda_{\max}^k} = \lambda_{\max}^k \frac{d}{d\alpha}\bigg|_{\alpha=0} \sum_{i \in M_{\lambda^k}/L} \Theta_i^2(\alpha).
$$

If $\min_{i \in M_{\lambda^1}} (\lambda_i^1 - \lambda_i^2) = 0$, $\lambda_i^1 - \lambda_i^2 < 0$ for any $i \in M_{\lambda^k}/L$, and thus, unless $\theta_i = 0$ for any $i \in M_{\lambda^k}/L$,

$$
\frac{d}{d\alpha}\bigg|_{\alpha=0} \sum_{i \in M_{\lambda^k}/L} \Theta_i^2(\alpha) = - \sum_{i \in M_{\lambda^k}/L} \alpha(\lambda_i^1 - \lambda_i^2)\theta_i^2 > 0
$$

.

On the other hand, if $\max_{i \in M_{\lambda^1}} (\lambda_i^1 - \lambda_i^2) = 0$, $\lambda_i^1 - \lambda_i^2 > 0$ for any $i \in M_{\lambda^k}/L$, and thus, unless $\theta_i = 0$ for any $i \in M_{\lambda^k}/L$,

$$
\frac{d}{d\alpha}\bigg|_{\alpha=0} \sum_{i \in M_{\lambda^k}/L} \Theta_i^2(\alpha) < 0.
$$

Therefore, in either case, the second claim holds.

Let $\theta$ be a first-order KKT point of Eq.21 with a KKT multiplier $\mu \in \mathbb{R}^n$. The stationary condition along the one-parameter family $\Theta(\alpha)$ is

$$
\frac{d}{d\alpha}\bigg|_{\alpha=0} \frac{1}{2} \|\Theta(\alpha)\|_{\Lambda_{\max}^k}^2 + \sum_{j \in [n]} \mu_j y_j \frac{d}{d\alpha}\bigg|_{\alpha=0} f(x_j; \Theta(\alpha)) = 0.
$$

Hence by the two properties above with both $k = 1, 2$, we obtain $\theta_i = 0$ for any $i \in (M_{\lambda^1} \cup M_{\lambda^2})/L$. Therefore $\theta$ is a first-order KKT point of Eq.22. $\qquad\square$

By exploiting all of the lemmas above, we are now going to prove Theorem B.1.

*proof of Theorem B.1.* Suppose there exists two different proper points $\lambda^1, \lambda^2 \in S$. By Lemma B.5, at least either of $\lambda^1$ or $\lambda^2$ is on the boundary $\partial S$ of $S$. Without loss of generality, we assume that $\lambda^1 \in \partial S$. We consider a line segment in $S$

$$
L := \{y(t) \in \mathbb{R}_{\geq 0}^d : y(t) = \zeta t + \lambda^1, t > 0\},
\tag{23}
$$

where $\zeta = \lambda^2 - \lambda^1 \in \mathbb{R}^d$.

We first consider the case where $L$ has a single end point $\lambda^1$. We will show that $\lambda^1$ and $\lambda^2$ are equivalent in terms of the resulting optimization problem, by applying Lemma B.7. Hence we are going to verify each condition required in the lemma.

Notice that the fact that $L$ has a single end point implies that $\zeta_i \geq 0$ for any $i \in [d]$. Combined with Lemma B.6, this means that

$$\min_{i \in M_{\lambda^k}} (\lambda_i^2 - \lambda_i^1) = 0$$

for $k = 1, 2$. Furthermore, clearly $\{i \in M_{\lambda^2} : \lambda_i^1 = \lambda_i^2\} = M_{\lambda^1}$. This is because for any $i \in \{i \in M_{\lambda^2} : \lambda_i^1 = \lambda_i^2\}$ and $j \in [d]$,

$$\lambda_i^1 = \lambda_i^2 \geq \lambda_j^2 \geq \lambda_j^2 - \zeta_j t = \lambda_j^1,$$

where the inequality above hold as an equality if and only if $j \in \{i \in M_{\lambda^2} : \lambda_i^1 = \lambda_i^2\}$.

$$\{i \in M_{\lambda^2} : \lambda_i^1 = \lambda_i^2\} = \{i \in M_{\lambda^1} : \lambda_i^1 = \lambda_i^2\}$$

immediately follows from $\{i \in M_{\lambda^2} : \lambda_i^1 = \lambda_i^2\} = M_{\lambda^1}$. Hence by applying Lemma B.7, we obtain that $\lambda^1$ and $\lambda^2$ are equivalent.

Next, we consider the other case where $L$ has two end points. Let $y(t_{\max})$ denote the other end point. Suppose $\lambda^2$ is an interior point, i.e., $\lambda^2 \neq y(t_{\max})$. By exploiting Lemma B.3, we obtain

$$\lambda_{\max}^1 = t_{\max} \max_{i \in J} \zeta_i > (t_{\max} - t^2) \max_{i \in J} \zeta_i = \lambda_{\max}^2, \tag{24}$$

where $t^2$ is given by $y(t^2) = \lambda^2$ and $J \subset [d]$ is given by

$$J := \{j \in [d] : y(t_{\max})_j = 0, \exists \lambda \in L, \lambda_j > 0\}.$$

On the other hand, by applying Lemma B.6 at $\lambda^1$, we know that there exists $i \in M_{\lambda^1}$ such that $\zeta \geq 0$. Hence $\lambda_i^2 \geq \lambda_i^1 = \lambda_{\max}^1$. This clearly contradicts with Eq.24. Hence, $\lambda^2 = y(t^{\max})$.

Lastly, we show that $\lambda^1$ and $\lambda^2 = y(t^{\max})$ are equivalent in terms of the resulting optimization problem by applying Lemma B.7. Hence, we are going to verify the conditions required in the lemma. Since $\min_{i \in M_{\lambda^2}} \zeta_i \leq 0$ by Lemma B.6, there exists $j \in M_{\lambda^2}$ such that $\zeta_j \leq 0$, and hence $\lambda_{\max}^2 = \lambda_j^2 = \lambda_j^1 + \zeta_j t_{\max} < \lambda_j^1 \leq \lambda_{\max}^1$. By applying Lemma B.6 at $\lambda^1$, similarly we obtain $\lambda_{\max}^1 \leq \lambda_{\max}^2$. Therefore

$$\lambda_{\max}^1 = \lambda_{\max}^2.$$

Suppose there exists $i \in M_{\lambda^1}$ such that $\zeta_i > 0$. Then, $\lambda_{\max}^1 = \lambda_i^1 = \lambda_i^2 - \zeta_i t_{\max} = \lambda_i^2 < \lambda_{\max}^2$. This is a contradiction, and hence $\max_{i \in M_{\lambda^1}} \zeta_i \leq 0$. Combining this with Lemma B.6, $\max_{i \in M_{\lambda^1}} \zeta_i = 0$. Similarly, suppose there exists $i \in M_{\lambda^2}$ such that $\zeta_i < 0$. Then, $\lambda_{\max}^2 = \lambda_i^2 = \lambda_i^1 + \zeta_i t_{\max} < \lambda_{\max}^1$. This is a contradiction, and hence $\min_{i \in M_{\lambda^2}} \zeta_i = 0$.

Let $k \in \{i \in M_{\lambda^1} : \lambda_i^1 = \lambda_i^2\}$. Then $\lambda_k^2 = \lambda_k^1 = \lambda_{\max}^1 = \lambda_{\max}^2$. Hence $k \in \{i \in M_{\lambda^2} : \lambda_i^1 = \lambda_i^2\}$. Similarly if $k \in \{i \in M_{\lambda^2} : \lambda_i^1 = \lambda_i^2\}$, $\lambda_k^1 = \lambda_k^2 = \lambda_{\max}^2 = \lambda_{\max}^1$. Hence $k \in \{i \in M_{\lambda^1} : \lambda_i^1 = \lambda_i^2\}$. Therefore

$$\{i \in M_{\lambda^1} : \lambda_i^1 = \lambda_i^2\} = \{i \in M_{\lambda^2} : \lambda_i^1 = \lambda_i^2\}.$$

Now, all the conditions in Lemma B.7 are satisfied, and hence $\lambda^1$ and $\lambda^2$ are equivalent. In summary, regardless of the finiteness of the line $L$, $\lambda^1$ and $\lambda^2$ are equivalent in terms of the first-order KKT condition. $\qquad\square$

## C LIMITATIONS OF OUR CURRENT ASSUMPTIONS

**A1.** While, many modern neural network architectures are $\Lambda$-quasi-homogeneous, this class of models cannot represent models with non-homogeneous activations such as the hyperbolic tangent function or the sigmoid function. While these activation functions are less common in practice, it would be interesting if there exists a direction towards generalizing our analysis to models using these functions. This could be of interest to the computational biology community as these monotonic activations that saturate are much more biologically-plausible then non-saturating homogeneous activations. One route could be studying the properties of *homothetic functions*, which are monotonic transformations of homogeneous functions. This class of functions has the same ordinal properties of homogeneous functions and is used extensively in economics Simon et al. (1994).

**A2 - A5.** These assumptions are equivalent, up to order, to the assumptions presented in Lyu & Li (2019) and are all quite standard in the literature studying the maximum margin bias of gradient descent. The strongest of these assumptions is A4, which assumes the training dynamics of the model are governed by the first-order ODE gradient flow. As discussed in section 7, an important future step would be to generalize our results to stochastic gradient descent (SGD). It is very possible that the hyperparameters of SGD, such as the learning rate and batch size, play an important role in determining the forces driving the limiting dynamics.

**A6.** This assumption implies the convergence of the decision boundary and is equivalent to directional convergence for a homogeneous model. This assumption is necessary to show that the model's prediction is bounded on the normalized training trajectory (Lemma E.1) and for a technical reason to show the alignment of $\frac{d\theta}{dt}$ and $\Lambda\theta$ (Lemma E.4). While a necessary assumption, we expect that this assumption can be weakened by exploiting the argument in Ji & Telgarsky (2020), which was applied for homogeneous functions. This could be an important step for future work as it is possible to construct settings where gradient flow will violate this assumption. See Appendix J of Lyu & Li (2019) for an example of a smooth homogeneous function where the limiting dynamics of gradient flow provably don't converge after normalization, but move along a circle. Another, more practical example, occurs for models where the residual block diverges. Because these parameters necessarily have a $\lambda$ value of $0$, then the normalized parameter $\hat{\theta}$ diverge as well. The divergence of a residual block essentially means that the skip connection of the model is effectively negligible and does not play an important role in the classification. Thus, we believe that if the skip connections play an important role for the classification task, then residual block does not diverge, and this assumption is satisfied. Additionally, like A7, this assumption restricts the possible proper values of $\Lambda$ as discussed in App. B.

**A7.** This assumption implies that $\lambda_{\max}$ parameters play a role in the classification task. This is the strongest of the assumptions we introduce in our work. This assumption is a restriction on the interaction of the parameterization of a model and the dataset, and thus it is difficult to assert its validity for general settings, without directly modeling the data as in section 5. That said, there are some standard settings where we can assert that this assumption is always true. First, it is trivially true that for a homogeneous model where $\|\hat{\theta}\|_{\Lambda_{\max}} = \|\hat{\theta}\|$ that A7 is always true. Additionally, for all models with batch normalization on the last hidden layer, such as a ResNet-18 model, then A7 is also true, as the last layer parameters are $\Lambda_{\max}$ parameters, and thus necessary for classification. However, there are other settings where it is less clear that A7 is satisfied. For example, for a fully-connected network with bias parameters the validity of A7 will depend on the data. This limitation is why we introduced Conjecture 5.1, which does not involve A7, and provided empirical evidence supporting its claim in section 5. The challenge to proving Conjecture 5.1 will be showing that a version of Lemma 4.1 still holds once the parameter space is restricted such that the $\Lambda_{\max}$ parameters are zero. It is likely the case that an argument in this direction will require a proof by induction on the highest-rate parameters and their importance to the classification task.

## D   PROOF OF LEMMA 4.1

Throughout this section, we assume A1 to A4. To simplify notation in our proof, we define

$$\nu := \frac{1}{2}\frac{d}{dt}\|\theta\|_\Lambda^2. \tag{25}$$

We make use of the following two simple statements:

**Lemma D.1.** *The derivative of loss is given by*

$$\frac{d\mathcal{L}}{dt} = -\left\|\frac{d\theta}{dt}\right\|^2, \tag{26}$$

*for almost every $t > 0$. Hence, it is non-increasing for $t > 0$.*

*Proof.* Since $\mathcal{L}$ is locally Lipschitz function admitting a chain rule, by applying Lemma 5.2 in Davis et al. (2020) to $\mathcal{L}(\theta(t))$, we immediately obtain $\frac{d\mathcal{L}}{dt} = -\left\|\frac{d\theta}{dt}\right\|^2$ for almost every $t > 0$. □

**Lemma D.2.** *For all $\theta \in \mathbb{R}^m$, $\|\Lambda\theta\|^2 \le \lambda_{\max}\|\theta\|_\Lambda^2$.*

*Proof.* $\|\Lambda\theta\|^2 = \sum \lambda_i^2\theta_i^2 \le \lambda_{\max}\sum \lambda_i\theta_i^2 = \lambda_{\max}\|\theta\|_\Lambda^2$. □

We will now prove the main lemmas described in section 4.

*Proof of Lemma 4.1.* We first prove $\nu \ge \mathcal{L}\log((n\mathcal{L})^{-1})$. By Lemma A.3, we can express $\nu$ as

$$\nu = \left\langle \frac{d\theta}{dt}, \Lambda\theta \right\rangle = n^{-1}\sum_a e^{-y_a f_a(\theta)}y_a f_a(\theta).$$

Using this equality the statement can be shown by the following inequality

$$\mathcal{L}^{-1}\nu - \log(n\mathcal{L})^{-1} = \left(\sum_a e^{-y_a f_a(\theta)}\right)^{-1}\sum_a y_a f_a(\theta)e^{-y_a f_a(\theta)} - \log(n\mathcal{L})^{-1}$$

$$= -\sum_a p_a\log p_a \ge 0,$$

where $p_a := \left(\sum_a e^{-y_a f_a(\theta)}\right)^{-1}e^{-y_a f_a(\theta)}$. We now prove $\frac{d}{dt}\log(\tilde{\gamma}) \ge \lambda_{\max}^{-1}\frac{d}{dt}\log(\|\theta\|_\Lambda)\tan(\omega)^2$.

$$\frac{d}{dt}\log(\tilde{\gamma}) = \frac{d}{dt}\left(\log\log(n\mathcal{L})^{-1} - \lambda_{\max}^{-1}\log\|\theta\|_\Lambda\right)$$

$$= (\log(n\mathcal{L})^{-1})^{-1}\mathcal{L}^{-1}\left(-\frac{d\mathcal{L}}{dt}\right) - \frac{\langle\Lambda\theta,\frac{d\theta}{dt}\rangle}{\lambda_{\max}\|\theta\|_\Lambda^2}$$

$$\ge \nu^{-1}\left(\left(-\frac{d\mathcal{L}}{dt}\right) - \frac{\langle\Lambda\theta,\frac{d\theta}{dt}\rangle^2}{\|\Lambda\theta\|^2}\right),$$

where the last inequality applied Lemma 4.1 and Lemma D.2. Since $-\frac{d\mathcal{L}}{dt} = \left\|\frac{d\theta}{dt}\right\|^2$ for almost every $t > 0$, we can further simplify the RHS,

$$= \nu^{-1}\left(\left\|\frac{d\theta}{dt}\right\|^2 - \frac{\langle\Lambda\theta,\frac{d\theta}{dt}\rangle^2}{\|\Lambda\theta\|^2}\right)$$

$$= \nu^{-1}\left\|\left(I - \frac{\Lambda\theta\theta^\mathsf{T}\Lambda}{\|\Lambda\theta\|^2}\right)\frac{d\theta}{dt}\right\|^2$$

$$= \frac{\|v\|^2}{\nu}\frac{\|u\|^2}{\|v\|^2}$$

$$= \frac{\nu}{\|\Lambda\theta\|^2}\tan(\omega)^2,$$

where the *normal* component $v$ and the *tangent* component $u$ are defined as,

$$v := \left( \frac{\Lambda \theta \theta^\intercal \Lambda}{\|\Lambda \theta\|^2} \right) \frac{d\theta}{dt}, \qquad u := \left( I - \frac{\Lambda \theta \theta^\intercal \Lambda}{\|\Lambda \theta\|^2} \right) \frac{d\theta}{dt}, \tag{27}$$

and the last equality used $\frac{\|v\|^2}{\nu} = \frac{\nu}{\|\Lambda \theta\|^2}$ and the definition of $\omega$. Applying Lemma D.2 and $\frac{d}{dt} \log(\|\theta\|_\Lambda) = \frac{\nu}{\|\theta\|_\Lambda^2}$ gives the final result,

$$\frac{d}{dt} \log(\tilde{\gamma}) \geq \lambda_{\max}^{-1} \frac{\nu}{\|\theta\|_\Lambda^2} \tan(\omega)^2 = \lambda_{\max}^{-1} \frac{d}{dt} \log(\|\theta\|_\Lambda) \tan(\omega)^2.$$

$\square$

Here is an important direct consequence of Lemma 4.1.

**Corollary D.1.** $\tilde{\gamma}(t)$ *is non-decreasing for* $t \geq t_0$.

*Proof.* Notice the following inequality derived using Lemma 4.1, Lemma D.1, and A5,

$$\frac{d}{dt} \log(\|\theta\|_\Lambda) = \frac{\nu}{\|\theta\|_\Lambda^2} \geq \frac{\mathcal{L} \log((n\mathcal{L})^{-1})}{\|\theta\|_\Lambda^2} \geq 0.$$

It follows by Lemma 4.1 that $\frac{d}{dt} \log(\tilde{\gamma}) \geq 0$ and thus $\tilde{\gamma}$ is non-decreasing. $\square$

# E  PROOF OF THEOREM 4.1

We first prove the following four lemmas by utilizing Lemma 4.1.

**Lemma E.1** (Bounded $f(x;\theta)$ on $\hat{\theta}(t)$). *Under assumptions A1, A2, A4, and A6, then for all $i \in [n]$ $f(x_i;\theta)$ is bounded on the normalized training trajectory $\hat{\theta}(t)$ (i.e. $\exists k > 0$ such that $|f(x_i;\hat{\theta}(t))| \leq k$ for all $t \geq 0$ and all $i \in [n]$).*

**Lemma E.2** (Divergence of $\mathcal{L}^{-1}, q_{\min}, \|\theta\|_\Lambda^2$). *Under assumptions A1 to A5, as $t \to \infty$ the quantities $\mathcal{L}^{-1}, q_{\min}, \|\theta\|_\Lambda^2 \to \infty$.*

**Lemma E.3** (Upper bound on $\tilde{\gamma}$). *Under assumptions A1 to A7, the normalized margin $\tilde{\gamma}$ is bounded above by a constant.*

**Lemma E.4** (Alignment of $\frac{d\theta}{dt}$ and $\Lambda\theta$). *Under assumptions A1 to A7, there exists a sequence of time $\{t_k\}_{k\in\mathbb{N}}$ on which cosine similarity $\beta(t_k) \to 1$.*

***Proof of Lemma E.1***.  By A4 and the uniqueness of the normalization procedure (Lemma A.1) we know the normalized training trajectory $\hat{\theta}(t)$ is continuous. Combined with the convergence of the normalized parameters (A6), this implies that $\{\hat{\theta}(t) : t \geq 0\}$ is bounded. Which, by the continuity of $f$, further implies that for any fixed $x \in \mathbb{R}^d$, $\{f(x;\hat{\theta}(t)) : t \geq 0\}$ is bounded. Thus, there exists a $k > 0$ such that for all $i \in [n]$, $|f(x_i;\hat{\theta}(t))| \leq k$ for all $t \geq 0$. $\qquad\square$

***Proof of Lemma E.2***.  We first prove that $\mathcal{L}^{-1} \to \infty$ as $t \to \infty$. By Lemma D.1,

$$-\frac{d\mathcal{L}}{dt} = \left\|\frac{d\theta}{dt}\right\|^2 \geq \left\|\left(\frac{\Lambda\theta\theta^\mathsf{T}\Lambda}{\|\Lambda\theta\|^2}\right)\frac{d\theta}{dt}\right\|^2 \geq \lambda_{\max}^{-1}\frac{\nu^2}{\|\theta\|_\Lambda^2},$$

where the first equality holds in almost everywhere sense, and for the last inequality we use the definition of $\nu$ and Lemma D.2. Applying Lemma 4.1, we can lower bound $\nu$, giving the lower bound

$$-\frac{d\mathcal{L}}{dt} \geq \lambda_{\max}^{-1}\frac{\mathcal{L}^2 \log((n\mathcal{L})^{-1})^2}{\|\theta\|_\Lambda^2}$$

$$= \lambda_{\max}^{-1}\mathcal{L}^2 \log((n\mathcal{L})^{-1})^{(2-2\lambda_{\max})}\left(\frac{\log((n\mathcal{L})^{-1})}{\|\theta\|_\Lambda^{\lambda_{\max}^{-1}}}\right)^{2\lambda_{\max}}$$

$$\geq \lambda_{\max}^{-1}\mathcal{L}^2 \log((n\mathcal{L})^{-1})^{(2-2\lambda_{\max})}\tilde{\gamma}(t_0)^{2\lambda_{\max}}.$$

where the last inequality holds almost everywhere sense via Corollary D.1. Rearranging terms on both sides of the inequality gives,

$$-\frac{d\mathcal{L}}{dt}\mathcal{L}^{-2}\log((n\mathcal{L})^{-1})^{-2(1-\lambda_{\max})} \geq \lambda_{\max}^{-1}\tilde{\gamma}(t_0)^{2\lambda_{\max}}.$$

Integration from $t_0$ to $t$, with the substitution $-\frac{d\mathcal{L}}{dt}\mathcal{L}^{-2} = \frac{d}{dt}\mathcal{L}^{-1}$, gives

$$\int_{\mathcal{L}^{-1}(t_0)}^{\mathcal{L}^{-1}(t)}(\log n^{-1}z)^{-2(1-\lambda_{\max})}dz \geq \lambda_{\max}^{-1}\tilde{\gamma}(t_0)^{2\lambda_{\max}}(t-t_0).$$

The RHS diverges as $t \to \infty$, and the LHS as a function of $t$ is non-decreasing for $z \geq n$, which is true for all $t \geq t_0$ by Lemma D.1 and A5. Thus we can conclude that $\mathcal{L}^{-1} \to \infty$. This implies $q_{\min} \to \infty$ as $t \to \infty$, because $q_{\min}$ is lower bounded by

$$\log(\mathcal{L}^{-1}) \leq q_{\min}.$$

We now show $\|\theta\|_\Lambda \to \infty$ as $t \to \infty$. We can upper bound $q_{\min}$ by

$$q_{\min} = e^{\tau(\theta)}\hat{q}_{\min} \leq e^{\tau_{\max}(\|\theta\|_\Lambda)}\sup_{i\in[n],t\geq 0}|f(x_i;\hat{\theta}(t))|,$$

where $\tau_{\max}(\rho)$ is defined as

$$\tau_{\max}(\rho) = \max\{\tau(\theta) : \|\theta\|_\Lambda = \rho\} = \lambda_{\min^+}^{-1}\log\rho,$$

and $\lambda_{\min^+} = \min_{\lambda_i > 0} \lambda_i$. By Lemma E.1, $\sup_{i \in [n], t \geq 0} |f(x_i; \hat{\theta}(t))| \leq k$ implying

$$\left(\frac{q_{\min}}{k}\right)^{\lambda_{\min^+}} \leq \|\theta\|_\Lambda,$$

and thus $\|\theta\|_\Lambda \to \infty$ as $t \to \infty$. $\qquad\square$

***Proof of Lemma E.3.*** We prove by construction a constant upper bound of $\tilde{\gamma}$. Notice, $\|\theta\|_{\Lambda_{\max}}^2 \leq \|\theta\|_\Lambda^2$, and thus we can easily upper bound $\tilde{\gamma}$ by

$$\tilde{\gamma} = \frac{\log((n\mathcal{L})^{-1})}{\|\theta\|_\Lambda^{\lambda_{\max}^{-1}}} \leq \frac{\log(\mathcal{L}^{-1})}{\|\theta\|_{\Lambda_{\max}}^{\lambda_{\max}^{-1}}}.$$

Notice that $\|\theta\|_{\Lambda_{\max}}^{\lambda_{\max}^{-1}} = e^\tau \|\hat{\theta}\|_{\Lambda_{\max}}^{\lambda_{\max}^{-1}}$ and by A7, we know that $\|\hat{\theta}\|_{\Lambda_{\max}} \geq \kappa$. Therefore, we can further upper bound $\tilde{\gamma}$ as

$$\tilde{\gamma} \leq \frac{\log(\mathcal{L}^{-1})}{e^\tau \kappa^{\lambda_{\max}^{-1}}} \leq \frac{e^{-\tau} q_{\min}}{\kappa^{\lambda_{\max}^{-1}}}$$

where the last inequality used $\log(\mathcal{L}^{-1}) \leq q_{\min}$. By Lemma E.1, $e^{-\tau} q_{\min} \leq \sup_{i \in [n], t \geq 0} |f(x_i; \hat{\theta}(t))| \leq k$ and therefore $\tilde{\gamma}$ is upper bounded by a constant,

$$\tilde{\gamma} \leq \frac{k}{\kappa^{\lambda_{\max}^{-1}}}.$$

$\qquad\square$

***Proof of Lemma E.4.*** We will inductively construct a sequence $\{t_k\}_{k \in \mathbb{N}}$ such that $\beta(t_k) - 1 < k^{-1}$ for any $k \in \mathbb{N}$. Assume that a sequence satisfies this condition for any $k < l$ with a positive integer $l$. We show that we can find $t_l > t_{l-1}$ such that the conditions above are met at $t_l$ as well. By Lemma.E.3 and Corollary D.1, we can find $l \in \mathbb{N}$ such that $s > t_{l-1} + \epsilon$ and

$$\log \tilde{\gamma}_{\max}(\infty) - \log \tilde{\gamma}_{\max}(s) < l^{-1}.$$

Here $\epsilon > 0$ is a constant to make sure that $\{t_k\}_{k \in \mathbb{N}}$ goes to infinity.

Furthermore, by the fact that $\rho \to \infty$, we can find $s' > s$ such that

$$\log \rho(s') - \log \rho(s) = 1.$$

This choice of $s$ and $s'$ satisfies

$$D := \frac{\log \tilde{\gamma}_{\max}(s') - \log \tilde{\gamma}_{\max}(t_l)}{\log \rho(s') - \log \rho(s)} < l^{-1}.$$

Assume that for any $t \in (s, s')$, $\beta^{-2}(l) - 1 > D$. Then

$$\log \tilde{\gamma}_{\max}(s') - \log \tilde{\gamma}_{\max}(s) < \int_s^{s'} (\beta^{-2} - 1) \frac{d \log \rho}{dt} dt.$$

On the other hand, by Lemma.4.1, the right hand side can be upper bounded as follows.

$$\int_{s'}^s (\beta^{-2} - 1) \frac{d \log \rho}{dt} dt \leq \int_{s'}^s \frac{d \log \tilde{\gamma}_{\max}}{dt} dt = \log \tilde{\gamma}_{\max}(s') - \log \tilde{\gamma}_{\max}(s).$$

This is a contradiction, implying that there exists $t \in (s, s')$ such that $\beta(s)^{-2} - 1 < D$. Thus,

$$|1 - \beta(t)| < 1 - 1/\sqrt{D+1} < 1 - 1/\sqrt{l^{-1} + 1} < l^{-1},$$

i.e., $t_l = t$ satisfies the condition. Note that $\lim_{l \to \infty} t_l \to \infty$ since $t_l \geq s > t_{l-1} + \epsilon$ for any $l$. $\quad\square$

Equipped with these convergence results, we can now prove Theorem 4.1 by exploiting the argument on the approximate KKT condition introduced in Dutta et al. (2013). In this paper, they defined a notion of $(\epsilon, \delta)$-KKT point, which can be stated for our optimization problem P as follows:

A point $\theta \in \mathbb{R}^m$ is a first-order $(\epsilon, \delta)$-KKT Point of P if there exist multipliers $\mu = (\mu_1, \ldots, \mu_n)$ such that the following four conditions are satisfied:

1. *Primal Feasibility:* $y_i f(x_i; \theta) \geq 1$ for all $i \in [n]$.

2. *Dual Feasability:* $\mu_i \geq 0$ for all $i \in [n]$

3. *Approximate Stationarity:*
   $\left\| \nabla_\theta \frac{1}{2} \|\theta\|_{\Lambda_{max}}^2 - \sum_i \mu_i y_i h_i \right\| \leq \epsilon$ with some $h_i \in \partial_\theta^\circ f(x_i; \theta)$ for each $i \in [n]$.

4. *Approximate Complementarity:* $\sum_i \mu_i \left( y_i f(x_i; \theta) - 1 \right) \leq \delta$.

We call this set of four conditions as the first-order $(\epsilon, \delta)$-KKT condition. In the proof for Theorem 4.1, we use the following fact.

**Lemma E.5.** *Under assumptions A1 to A5 and A7, $\psi_\alpha(\theta(t))$ with $\alpha = -\log(q_{\min})$ satisfies the first-order $(\epsilon(t), \delta(t))$-KKT condition with a multiplier $\mu(t)$, where $\epsilon(t), \delta(t), \mu(t)$ are given as follows:*

$$
\begin{cases}
\epsilon(t) = \sqrt{\lambda_{\max}} \tilde{\gamma}^{-\lambda_{\max}} \left( 2(1-\beta) + m \max_{i \in [m]: \lambda_i \neq \lambda_{\max}} q_{\min}^{2(\lambda_i - \lambda_{\max})} \right)^{1/2} \\
\delta(t) = e^{-1} n \lambda_{\max} \tilde{\gamma}^{-2\lambda_{\max}} \log((n\mathcal{L})^{-1})^{-1} \\
\mu_i(t) = c^{-1} q_{\min}^{(1-2\lambda_{\max})} e^{-q_i} \quad \forall i \in [m].
\end{cases}
\tag{28}
$$

*Here $c = \|\frac{d\theta}{dt}\| / \|\Lambda\theta\|$.*

We prove this lemma right after showing the proof of Theorem 4.1.

***Proof of Theorem 4.1.*** By Corollary of Theorem 3.6 in Dutta et al. (2013) (or Theorem C.4 in Lyu & Li (2019)), it suffices to show the following two statements:

- There exist a sequence of time $\{t_k\}_{k \in \mathbb{N}}$ and a sequence of real numbers $\{\alpha_k\}_{k \in \mathbb{N}}$ such that $\psi_{\alpha_k}(\theta(t_k))$ satisfies the first-order $(\epsilon_k, \delta_k)$-KKT condition, where $\epsilon_k, \delta_k \to 0$ and $\psi_{\alpha_k}(\theta(t_k))$ converges.

- $\lim_{k \to \infty} \psi_{\alpha_k}(\theta(t_k))$ satisfies MFCQ condition.

The second point directly follows from Lemma A.4. We prove the first statement with the result of Lemma E.5. By Lemma E.4, we can find a sequence of time $\{t_k\}_{k \in \mathbb{N}}$ on which $\beta \to 1$. Furthermore, because $\tilde{\gamma}$ is lower bounded by Lemma E.3, $q_{\min} \to \infty$ by Lemma E.2 and $\mathcal{L} \to 0$ by Lemma E.2, $\epsilon(t), \delta(t)$ on this sequence converge to 0: $\epsilon(t_k), \delta(t_k) \to 0$. Lastly we show $\psi_{\alpha(t_k)}(\theta(t_k))$ converges. Notice that $\psi_{\alpha(t_k)}(\theta(t_k)) = \hat{q}_{\min}^{-\Lambda}(t_k) \hat{\theta}(t_k)$, where $\hat{q}_{\min}(t_k)$ is the minimum margin of the model $f(x; \hat{\theta}(t_k))$. By A6, $\hat{\theta}(t_k)$ converges, and by the continuity of $f(x; \theta)$, $\hat{q}_{\min}(t_k)$ converges. Therefore to show the convergence of $\psi_{\alpha(t_k)}(\theta(t_k))$, it suffices to show that $\hat{q}_{\min}(t)$ is lower-bounded by a positive value. This can be seen as follows:

$$
\hat{q}_{\min} = \left( \frac{\|\hat{\theta}\|_{\Lambda_{max}}}{\|\theta\|_{\Lambda_{max}}} \right)^{\lambda_{\max}^{-1}} \quad q_{\min} \geq \frac{q_{\min}}{\|\theta\|_{\Lambda_{max}}^{\lambda_{\max}^{-1}}} \geq \frac{\log \mathcal{L}^{-1}}{\|\theta\|_{\Lambda_{max}}^{\lambda_{\max}^{-1}}} = \tilde{\gamma},
$$

where $\tilde{\gamma}$ is non-decreasing and upper-bounded by Corollary D.1. $\qquad \square$

***Proof of Lemma E.5.*** We verify each of four conditions one by one.

*1. Primal Feasibility:*
It is straight forward to check that this choice of $\alpha$ implies $\psi_\alpha(\theta)$ satisfies primal feasibility, as for all $i \in [n]$,

$$
y_i f(x_i; \psi_\alpha(\theta)) = e^\alpha y_i f(x_i; \theta) \geq e^\alpha q_{\min} = 1.
$$

*2. Dual Feasibility:*
It is clear that this choice of $\mu$ satisfies dual feasibility as $q_{\min} > 0$.

*3. Approximate Stationarity:*
By Lemma A.2, for any $h \in \partial_\theta^\circ f(x_i, \theta)$, we can expand the sum as follows:

$$\sum_i \mu_i y_i \partial_\theta^\circ f(x_i; \psi_\alpha(\theta)) = \sum_i \mu_i y_i e^{\alpha(I-\Lambda)} \partial_\theta^\circ f(x_i; \theta)$$

$$= c^{-1} q_{\min}^{(1-2\lambda_{\max})} q_{\min}^{(-I+\Lambda)} \left( \sum_i e^{-q_i} y_i \partial_\theta^\circ f(x_i; \theta) \right)$$

$$= c^{-1} q_{\min}^{(\Lambda-2\lambda_{\max})} \left( -\partial_\theta^\circ \mathcal{L} \right).$$

Hence, there exists a combination of $h_i \in \partial_\theta^\circ f(x_i, \psi_\alpha(\theta))$ such that $\sum_i \mu_i y_i h_i = c^{-1} q_{\min}^{(\Lambda-2\lambda_{\max})} \frac{d\theta}{dt}$.
By substituting the expression of $c$, we obtain

$$\sum_i \mu_i y_i h_i = q_{\min}^{-\lambda_{\max}} \|\Lambda\theta\| Q \frac{\frac{d\theta}{dt}}{\|\frac{d\theta}{dt}\|},$$

where $Q$ is a diagonal matrix such that $Q_{ii} = q_{\min}^{(\lambda_i - \lambda_{\max})}$. Now consider the derivative,

$$\nabla_\theta \frac{1}{2} \|\psi_\alpha(\theta)\|_{\Lambda_{\max}}^2 = D\Lambda\psi_\alpha(\theta) = q_{\min}^{-\lambda_{\max}} D\Lambda\theta,$$

where $D$ is a diagonal indicator matrix such that $D_{ii} = 1$ iff $\lambda_i = \lambda_{\max}$. Combining these last two expressions together, we can now bound the squared norm,

$$\left\| \nabla_\theta \frac{1}{2} \|\psi_\alpha(\theta)\|_{\Lambda_{\max}}^2 - \sum_i \mu_i y_i h_i \right\|^2 = \left\| q_{\min}^{-\lambda_{\max}} D\Lambda\theta - q_{\min}^{-\lambda_{\max}} \|\Lambda\theta\| Q \frac{\frac{d\theta}{dt}}{\|\frac{d\theta}{dt}\|} \right\|^2$$

$$= q_{\min}^{-2\lambda_{\max}} \|\Lambda\theta\|^2 \left\| D\frac{\Lambda\theta}{\|\Lambda\theta\|} - Q\frac{\frac{d\theta}{dt}}{\|\frac{d\theta}{dt}\|} \right\|^2$$

$$\leq \lambda_{\max} \left( \frac{q_{\min}}{\|\theta\|_\Lambda^{\lambda_{\max}^{-1}}} \right)^{-2\lambda_{\max}} \left\| D\frac{\Lambda\theta}{\|\Lambda\theta\|} - Q\frac{\frac{d\theta}{dt}}{\|\frac{d\theta}{dt}\|} \right\|^2$$

$$\leq \lambda_{\max} \tilde{\gamma}^{-2\lambda_{\max}} \left\| D\frac{\Lambda\theta}{\|\Lambda\theta\|} - Q\frac{\frac{d\theta}{dt}}{\|\frac{d\theta}{dt}\|} \right\|^2,$$

where the second to last inequality applies Lemma D.2, and the last inequality applies the definition of the normalized margin. We now bound the squared norm in the RHS using the definition of $\beta$,

$$\left\| D\frac{\Lambda\theta}{\|\Lambda\theta\|} - Q\frac{\frac{d\theta}{dt}}{\|\frac{d\theta}{dt}\|} \right\|^2 = \left\| D\left( \frac{\Lambda\theta}{\|\Lambda\theta\|} - \frac{\frac{d\theta}{dt}}{\|\frac{d\theta}{dt}\|} \right) + (D-Q)\frac{\frac{d\theta}{dt}}{\|\frac{d\theta}{dt}\|} \right\|^2$$

$$\leq \left\| D\left( \frac{\Lambda\theta}{\|\Lambda\theta\|} - \frac{\frac{d\theta}{dt}}{\|\frac{d\theta}{dt}\|} \right) \right\|^2 + \left\| (D-Q)\frac{\frac{d\theta}{dt}}{\|\frac{d\theta}{dt}\|} \right\|^2$$

$$\leq \left\| \frac{\Lambda\theta}{\|\Lambda\theta\|} - \frac{\frac{d\theta}{dt}}{\|\frac{d\theta}{dt}\|} \right\|^2 + \|D-Q\|^2$$

$$\leq \left\| \frac{\Lambda\theta}{\|\Lambda\theta\|} - \frac{\frac{d\theta}{dt}}{\|\frac{d\theta}{dt}\|} \right\|^2 + \sum_{\lambda_i \neq \lambda_{\max}} q_{\min}^{2(\lambda_i - \lambda_{\max})}$$

$$\leq 2(1-\beta) + m \max_{\lambda_i \neq \lambda_{\max}} q_{\min}^{2(\lambda_i - \lambda_{\max})}$$

Combing the upper bounds we get

$$\left\| \nabla_\theta \frac{1}{2} \|\psi_\alpha(\theta)\|_{\Lambda_{\max}}^2 - \sum_i \mu_i y_i h_i \right\|^2 \leq \lambda_{\max} \tilde{\gamma}^{-2\lambda_{\max}} \left( 2(1-\beta) + m \max_{\lambda_i \neq \lambda_{\max}} q_{\min}^{2(\lambda_i - \lambda_{\max})} \right)$$

$$= \epsilon^2(t).$$

*4. Approximate Complementary:*

Expand the following sum with the defined values for $\alpha$ and $\mu_i$,

$$\sum_i \mu_i \left(y_i f(x_i; \psi_\alpha(\theta)) - 1\right) = \sum_i \mu_i \left(q_{\min}^{-1} y_i f(x_i; \theta) - 1\right)$$

$$= \sum_i \frac{\mu_i}{q_{\min}} \left(q_i - q_{\min}\right)$$

$$= c^{-1} q_{\min}^{-2\lambda_{\max}} \sum_i e^{-q_i} \left(q_i - q_{\min}\right)$$

Notice the lower bound on the scalar $c$,

$$c = \frac{\|\frac{d\theta}{dt}\|}{\|\Lambda\theta\|} \geq \frac{|\langle \frac{d\theta}{dt}, \frac{\Lambda\theta}{\|\Lambda\theta\|} \rangle|}{\|\Lambda\theta\|} = \frac{\nu}{\|\Lambda\theta\|^2} \geq \frac{\mathcal{L} \log((n\mathcal{L})^{-1})}{\|\Lambda\theta\|^2} \geq \frac{e^{-q_{\min}} \log((n\mathcal{L})^{-1})}{\|\Lambda\theta\|^2}$$

where the last inequality follows from $\mathcal{L} \geq e^{-q_{\min}}$. Using this lower bound for $c$, we can upper bound the previous expression as

$$\sum_i \mu_i \left(y_i f(x_i; \psi_\alpha(\theta)) - 1\right) \leq q_{\min}^{-2\lambda_{\max}} \|\Lambda\theta\|^2 \log((n\mathcal{L})^{-1})^{-1} \left(\sum_i e^{-(q_i - q_{\min})} \left(q_i - q_{\min}\right)\right)$$

The function $f(z) = e^{-z} z$ attains its global maximum at $z = 1$, implying we can further upper bound this quantity as

$$\sum_i \mu_i \left(y_i f(x_i; \psi_\alpha(\theta)) - 1\right) \leq e^{-1} n q_{\min}^{-2\lambda_{\max}} \|\Lambda\theta\|^2 \log((n\mathcal{L})^{-1})^{-1}$$

$$\leq e^{-1} n \left(\frac{q_{\min}}{\|\theta\|_\Lambda^{\lambda_{\max}^{-1}}}\right)^{-2\lambda_{\max}} \log((n\mathcal{L})^{-1})^{-1} \lambda_{\max}$$

$$= e^{-1} n \tilde{\gamma}^{-2\lambda_{\max}} \log((n\mathcal{L})^{-1})^{-1} \lambda_{\max}$$

$$= \delta(t),$$

where the second to last inequality applies Lemma D.2. $\qquad\square$

# F   PROOF OF LEMMA 5.1

In this section, we provide a proof of Lemma 5.1.

*Proof of Lemma 5.1.* Notice that in the homogeneous case, all the parameters have the largest $\lambda$, and therefore $P = I$ and $P_\perp = 0$. By substituting these to the expressions of $w_{\text{quasi}-\text{hom}}$ and $l(w_{\text{quasi}-\text{hom}})$ in Eq.5.1, we can obtain those of $w_{\text{hom}}$ and $l(w_{\text{hom}})$. Therefore in the following, we focus on proving expressions for the quasi-homogeneous case.

By symmetry, Eq.4 can be reduced to the following optimization problem over a single ball,

$$\text{Eq.4} = \min_{w \in \mathbb{R}^d} \left[ \|Pw\| : \min_{x \in B(\mu,r)} \langle w, x \rangle \geq 1 \right].$$

The minimization over $x \in B(\mu, r)$ above can be further reduced as follows:

$$\begin{aligned}
\min_{x \in B(\mu,r)} \langle w, x \rangle &= \langle w, \mu \rangle - \max_{x \in B(0,r)} \langle w, x \rangle \\
&= \langle w, \mu \rangle - r\|w\| \\
&= \langle Pw, P\mu \rangle + \langle P_\perp w, P_\perp \mu \rangle - r\|w\|
\end{aligned}$$

.

Hence, the optimization problem Eq.4 can be reduced as follows:

$$\begin{aligned}
\text{Eq.4} &= \min_{w \in \mathbb{R}^d} \left[ \|Pw\| : \langle Pw, P\mu \rangle + \langle P_\perp w, P_\perp \mu \rangle - r\|w\| \geq 1 \right] \\
&= \min_{w_1 \in \mathbb{R}^m, w_2 \in \mathbb{R}^{d-m}} \left[ \|w_1\| : \langle w_1, P\mu \rangle + \langle w_2, P_\perp \mu \rangle - r\sqrt{\|w_1\|^2 + \|w_2\|^2} \geq 1 \right]
\end{aligned} \qquad (29)$$

Here we split the optimization over $w \in \mathbb{R}^d$ by considering the orthogonal vectors $w_1 = Pw$ and $w_2 = P_\perp w_2$ separately. Because the objective function $\|w_1\|$ is independent of $w_2$, the last expression above is equivalent to the following:

$$\min_{w_1 \in \mathbb{R}^m} \left[ \|w_1\| : \langle w_1, P\mu \rangle + \max_{w_2 \in \mathbb{R}^{d-m}} \left[ \langle w_2, P_\perp \mu \rangle - r\sqrt{\|w_1\|^2 + \|w_2\|^2} \right] \geq 1 \right].$$

The maximization over $w_2 \in \mathbb{R}^{m-d}$ is achieved if $w_2$ is parallel to $P_\perp \mu$, and hence, by letting $\rho_w$ denote $\|w_2\|$,

$$\begin{aligned}
&\max_{w_2 \in \mathbb{R}^{d-m}} \left[ \langle w_2, P_\perp \mu \rangle - r\sqrt{\|w_1\|^2 + \|w_2\|^2} \right] \\
&= \max_{\rho_w \in \mathbb{R}_{\geq 0}} \left[ \rho_\mu \rho_w - r\sqrt{\|w_1\|^2 + \rho_w^2} \right] \\
&= \rho_\mu \left( \frac{\rho_\mu/r}{\sqrt{1 - \rho_\mu^2/r^2}} \|w_1\| \right) - r\sqrt{\|w_1\|^2 + \frac{\rho_\mu^2/r^2}{1 - \rho_\mu^2/r^2}\|w_1\|^2} \\
&= -r\sqrt{1 - \rho_\mu^2/r^2}\|w_1\|,
\end{aligned} \qquad (30)$$

where the maximization over $\rho_\mu \in \mathbb{R}_{\geq 0}$ on the second line of the equation above is achieved if only if $\rho_\mu = \frac{\rho_\mu/r}{\sqrt{1-\rho_\mu^2/r^2}}\|w_1\|$, and hence the maximization over $w_2 \in \mathbb{R}^{d-m}$ on the first line is achieved if and only if

$$w_2 = \frac{\|w_1\|}{r\sqrt{1 - \rho_\mu^2/r^2}} P_\perp \mu. \qquad (31)$$

Therefore, by substituting Eq.30, we obtain

$$
\begin{aligned}
\text{Eq.4} &= \min_{w_1 \in \mathbb{R}^m} \left[ \|w_1\| : \|w_1\| \left( \langle \frac{w_1}{\|w_1\|}, P\mu \rangle - r\sqrt{1 - \rho_\mu^2/r^2} \right) \geq 1 \right] \\
&= \min_{w_1 \in \mathbb{R}^m} \left[ \left( \langle \frac{w_1}{\|w_1\|}, P\mu \rangle - r\sqrt{1 - \rho_\mu^2/r^2} \right)^{-1} : \langle \frac{w_1}{\|w_1\|}, P\mu \rangle - r\sqrt{1 - \rho_\mu^2/r^2} > 0 \right] \\
&= \left( \max_{w_1 \in \mathbb{R}^m} \left[ \langle \frac{w_1}{\|w_1\|}, P\mu \rangle - r\sqrt{1 - \rho_\mu^2/r^2} : \langle \frac{w_1}{\|w_1\|}, P\mu \rangle - r\sqrt{1 - \rho_\mu^2/r^2} > 0 \right] \right)^{-1} \\
&= \left( \|P\mu\| - r\sqrt{1 - \rho_\mu^2/r^2} \right)^{-1} \\
&= \left( \sqrt{1 - \rho_\mu^2} - r\sqrt{1 - \rho_\mu^2/r^2} \right)^{-1}.
\end{aligned}
$$

Note that this quantity is positive since $r < 1$ by assumption, and the minimization over $w \in \mathbb{R}^m$ is achieved if and only if $w_1$ is parallel to $P\mu$,

$$
w_1 \propto P\mu. \tag{32}
$$

Notice that the optimizers of Eq.29, and hence those of Eq.4, need to satisfy both Eq.32 and Eq.31. This means that the optimizer is unique and is given as follows:

$$
w_{\text{quasi-hom}} = \left( \sqrt{1 - \rho_\mu^2} - r\sqrt{1 - \rho_\mu^2/r^2} \right)^{-1} \left( \frac{1}{\sqrt{1 - \rho_\mu^2}} P\mu + \frac{1}{r\sqrt{1 - \rho_\mu^2/r^2}} P_\perp \mu \right)
$$

By normalizing this, we obtain the expression in Eq.5. At this minimizer $w_{\text{quasi-hom}}$, the robustness $l$ is obtained as

$$
\begin{aligned}
l(w_{\text{quasi-hom}}) &= \|w_{\text{quasi-hom}}\|^{-1} \min_{x \in B(\mu, A)} \langle w_{\text{quasi-hom}}, x \rangle \\
&= \|w_{\text{quasi-hom}}\|^{-1} \langle w_{\text{quasi-hom}}, \mu \rangle - r \\
&= \sqrt{1 - r^{-2}\rho_\mu^2} \sqrt{1 - \rho_\mu^2} + r^{-1}\rho_\mu^2 - r \\
&= \sqrt{1 - r^{-2}\rho_\mu^2} \left( \sqrt{1 - \rho_\mu^2} - \sqrt{r^2 - \rho_\mu^2} \right)
\end{aligned}
$$

$\square$

# G   PROOF OF THEOREM 6.1

We here first state the rigorous version of Theorem 6.1

**Theorem G.1** (Neural Collapse). *Under assumptions A8, A9, and $d \geq C$, any global optimum of the optimization problem Eq.8 satisfies Neural Collapse, i.e.,*

- *For any class $c \in [C]$, there exists a vector $\bar{h}_c$ such that $h_i = \bar{h}_c$ for any data $i \in [n]$ with $y_i = c$.*

- *The convex hull of $\{w_c\}_{c \in [C]}$ forms a regular $(C-1)$-simplex, centered at the origin.*

- *For any $c \in [C]$, $\bar{h}_c$ and $w_c$ are equivalent up to re-scaling.*

- $\mathrm{argmax}_{c \in [C]} \, w_c^T h + b_c = \mathrm{argmin}_{c \in [C]} \left\| h - \bar{h}_c \right\|$ *for any $h \in \mathbb{R}^d$, i.e., any feature vector is classified to the class $c$ with the nearest class mean $\bar{h}_c$.*

**Proof sketch for Theorem G.1**. To prove Theorem G.1, we study a sequence of three relaxed optimization problems, starting from Eq.8, and introduce five lemmas (Lemma G.1 to G.5) characterizing the minimizers of these relaxed problems. The optimization problem Eq.8 can be reduced to the following by A8,

$$\min_{(w,b,h)} \sum_{c \in [C]} |w_c|^2 + |b|^2 \text{ s.t. } \begin{cases} \min_{i \in [n]} q_i \geq 1 \\ \sum_j (h_i)_j = 0, \sum_j (h_i)_j^2 = 1 \quad \forall i \in [n], \end{cases} \tag{33}$$

where $q_i \in \mathbb{R}$ is defnied as

$$q_i := (w_{y_i})^T h_i + b_{y_i} - \max_{c \neq y_i} \left[ (w_c)^T h_i + b_c \right]. \tag{34}$$

The minimizers of this optimization problem are characterized in Lemma G.5. To prove Lemma G.5, we will consider the following further relaxed problem,

$$\min_{(w,b,h)} \sum_{c \in [C]} |w_c|^2 + |b|^2 \text{ s.t. } \begin{cases} \min_{i \in [n]} q_i \geq 1 \\ \sum_j (h_i)_j^2 = 1 \quad \forall i \in [n]. \end{cases} \tag{35}$$

This problem is analyzed in Lemma G.4. This optimization problem is equivalent to our last relaxed problem,

$$\min_{(w,b)} \sum_{c \in [C]} |w_c|^2 + |b|^2 \text{ s.t. } \max_{\{h_i\}_{i \in [n]}} \left\{ \min_{i \in [n]} q_i : \sum_j (h_i)_j^2 = 1 \quad \forall i \in [n] \right\} \geq 1. \tag{36}$$

The minimizers of this problem are analyzed in Lemma G.3, with the help of Lemma G.2, which analyzes a further relaxed problem and Lemma G.1, which analyzes the constraint in Eq.36. We will now introduce and prove Lemmas G.1 through G.5.

Let $H$ denote the set of $\{h_i\}_{i \in [n]}$ satisfying the constraints $\sum_j (h_i)_j^2 = 1$ for any $i \in [n]$, $\Delta_c$ denote the $(C-2)$-simplex formed by $\{w_{c'}\}_{c' \in [C]/\{c\}}$, and $\Delta$ denote the standard $(C-2)$-simplex.

**Lemma G.1.** *Under assumption A9, the following equality holds*

$$\max_{\{h_i\} \in H} \min_{i \in [n]} q_i = \min_{c \in [C]} L_c, \tag{37}$$

*with*

$$L_c := \min_{\alpha \in \Delta} \|w_c - w_c'(\alpha)\| + (b - b_c'(\alpha)), \tag{38}$$

*where $w_c'(\alpha) : \Delta \to \Delta_c$ and $b_c'(\alpha) : \Delta \to \mathbb{R}$ are defined as $w_c'(\alpha) := \sum_{c' \in [C]/\{c\}} \alpha_{i_c(c')} w_{c'}$ and $b_c'(\alpha) := \sum_{c' \in [C]/\{c\}} \alpha_{i_c(c')} b_{c'}$, where $i_c(\cdot) : [C]/\{c\} \to [C-1]$ is given by $i_c(c') = c'$ if $c' < c$ and $i_c(c') = c' - 1$ otherwise. Any maximizer of the quantity above is given by $h_i = h_{y_i}^*$ where*

$$h_c^* = \frac{w_c - w_c^*}{\|w_c - w_c^*\|}, \tag{39}$$

*with $w_c^* = w_c'(\alpha)$ with $\alpha$ minimizing Eq.38.*

*Proof.*

$$\max_{\{h_i\}\in H} \min_{i\in[n]} q_i \quad = \quad \min_{i\in[n]} \max_{h:\|h\|=1} \left[ (w_{y_i})^T h + b_{y_i} - \max_{c'\in[C]/\{y_i\}} \left[ (w_{c'})^T h + b_{c'} \right] \right]$$

$$= \quad \min_{c\in[C]} \max_{h:\|h\|=1} \left[ (w_c)^T h + b_c - \max_{c'\in[C]/\{c\}} \left[ (w_{c'})^T h + b_{c'} \right] \right],$$

where the maximization over $\{h_i\}_{i\in[n]}$ is achieved when

$$h_i \in \arg\max_{h:\|h\|=1} \left[ (w_{y_i})^T h + b_{y_i} - \max_{c'\in[C]/\{y_i\}} \left[ (w_{c'})^T h + b_{c'} \right] \right].$$

The quantity inside the parenthesis can be reduced as follows.

$$(w_c)^T h + b_c - \max_{c'\in[C]/\{c\}} \left[ (w_{c'})^T h + b_{c'} \right] \quad = \quad \min_{c'\in[C]/\{c\}} (w_c - w_{c'})^T h + (b - b_{c'})$$

$$= \quad \min_{\alpha\in\Delta} (w_c - w_c'(\alpha))^T h + (b - b_c'(\alpha)),$$

Therefore by the minimax theorem,

$$\max_{h:\|h\|=1} \left[ (w_c)^T h + b_c - \max_{c'\in[C]/\{c\}} \left[ (w_{c'})^T h + b_{c'} \right] \right]$$

$$= \quad \max_{h:\|h\|=1} \min_{\alpha\in\Delta} (w_c - w_c'(\alpha))^T h + (b - b_c'(\alpha))$$

$$= \quad \min_{\alpha\in\Delta} \max_{h:\|h\|=1} (w_c - w_c'(\alpha))^T h + (b - b_c'(\alpha))$$

$$= \quad \min_{\alpha\in\Delta} \|w_c - w_c'(\alpha)\| + (b - b_c'(\alpha))$$

$$= \quad L_c,$$

where the maximization over $h$ is achieved if and only if $h = \frac{w_c - w_c^*}{\|w_c - w_c^*\|}$. Hence Eq.37 holds. Clearly, the maximizer is $h_i = \frac{w_{y_i} - w_{y_i}^*}{\|w_{y_i} - w_{y_i}^*\|}$. $\qquad\square$

By Lemma G.1, the optimization problem Eq.36 is reduced to

$$\min_{(w,b)} \sum_c |w_c|^2 + |b|^2 \ \text{s.t.} \ \min_{c\in[C]} L_c \geq 1. \tag{40}$$

We will later show that this minimization is achieved when the convex hull of $\{w_c\}_{c\in[C]}$ forms a regular simplex. However, before dealing with Eq.40, we first solve the minimization of the following easier problem.

**Lemma G.2.** *Consider*

$$Z_c := \|w_c - w_c'(\bar{\alpha})\| + (b - b_c'(\bar{\alpha})), \tag{41}$$

*where $\bar{\alpha} := ((C-1)^{-1}, (C-1)^{-1}, \cdots, (C-1)^{-1})$ is the bary-center of simplex $\Delta$. The minimizer of the following optimization problem*

$$\min_{(\{w_c\}_{c\in[C]}, b)} \sum_c |w_c|^2 + |b|^2 \ \text{s.t.} \ \min_{c\in[C]} Z_c \geq 1, \tag{42}$$

*is achieved if and only if the following conditions are met:*

$$\begin{cases} \|w_c\| = \frac{C-1}{C} \\ \sum_{c\in[C]} w_c = 0 \\ b = 0. \end{cases} \tag{43}$$

*Furthermore, at these minimizers $Z_c = 1$ for any $c \in [C]$.*

*Proof.* Notice that the constraint of this optimization problem is translationally invariant, i.e., for any $\bar{w} \in \mathbb{R}^d$ and any $\{w_c\}_{c \in [C]}$ satisfying $\min_{c \in [C]} Z_c \geq 1$, $\{w_c + \bar{w}\}_{c \in [C]}$ also satisfies the constraint. Hence, the minimizer of Eq.42 should satisfy the stationary condition with respect to the derivative of $\bar{w}_i$ for all $i \in [d]$, i.e.,

$$\sum_{c \in [C]} w_c = 0. \tag{44}$$

In the following, we consider the optimization with this new constraint Eq.44. Next, we relax the constraint $\min_c Z_c \geq 1$ to $C^{-1} \sum_c Z_c \geq 1$ (clearly $C^{-1} \sum_c Z_c \geq \min_{c \in [C]} Z_c$). By the help of Eq.44, this averaged value is calculated as

$$
\begin{aligned}
C^{-1} \sum_{c \in [C]} Z_c &= C^{-1} \sum_{c \in [C]} [\|w_c - w_c'(\bar{\alpha})\| + (b_c - b_c'(\bar{\alpha}))] \\
&= C^{-1} \sum_{c \in [C]} \left[ \left\| w_c - \frac{-1}{C-1} w_c \right\| + \left( (b_c - (C-1)^{-1} \sum_{c' \in [C]/\{c\}} b_{c'}) \right) \right] \\
&= C^{-1} \sum_{c \in [C]} \left[ \frac{C}{C-1} \|w_c\| + \left( (b_c - (C-1)^{-1} \sum_{c' \in [C]/\{c\}} b_{c'}) \right) \right] \\
&= (C-1)^{-1} \sum_{c \in [C]} \|w_c\|.
\end{aligned}
$$

Hence, the relaxed problem can be stated as follows:

$$\min_{(w,b)} \sum_c \|w_c\|^2 + \|b\|^2 \quad \text{s.t.} \quad \sum_{c \in [C]} \|w_c\| \geq C - 1.$$

Clearly, this can be achieved if and only if $b = 0$ and $\|w_c\| = \frac{C-1}{C}$. Notice that this configuration satisfies $C^{-1} \sum_c Z_c = \min_{c \in [C]} Z_c$, and hence it is also the global optimum of the original problem Eq.42. Lastly, it is easy to see that these minimizers satisfies

$$Z_c = \frac{C}{C-1} \|w_c\| + \left( b - (C-1)^{-1} \sum_{c' \in [C]/\{c\}} b_{c'} \right) = 1.$$

for any $c \in [C]$. □

**Lemma G.3.** *Under assumptions A9 and $d \geq C$, the minimization Eq.36 is achieved if and only if the following conditions are met:*

$$
\begin{cases}
\text{The convex hull of } \{w_c\}_{c \in [C]} \text{ forms a regular } (C-1)\text{-simplex} \\
\|w_c\| = \frac{C-1}{C} \quad \forall c \in [C] \\
\sum_{c \in [C]} w_c = 0 \\
b = 0.
\end{cases} \tag{45}
$$

*Proof.* First we show that any point satisfying Eq.45 is a minimizer of Eq.36. Notice that the set of $(\{w_c\}_{c \in [C]}, b)$ satisfying Eq.45 is non-empty since $d \geq C$. All elements of this set satisfy $Z_c = L_c$ for any $c \in [C]$ by the first condition in Eq.45, and are clearly minimizers of Eq.42 by the other conditions in Eq.45 as shown in Lemma G.2. Thus, the elements satisfy $\min_{c \in [C]} L_c \geq 1$. For any $c \in [C]$, $Z_c \geq L_c$, and hence

$$\left\{ (\{w_c\}_{c \in [C]}, b) : \min_{c \in [C]} Z_c \geq 1 \right\} \supseteq \left\{ (\{w_c\}_{c \in [C]}, b) : \min_{c \in [C]} L_c \geq 1 \right\}.$$

Therefore, a point satisfying Eq.45 is a minimizer Eq.36.

Next we prove the inverse. We assume that $(\{w_c\}_{c \in [C]}, b)$ is a minimizer of Eq.36. This point must also be a minimizer of Eq.42, because as we showed previously, there exists a minimizer of Eq.42

satisfying $\min_{c \in [C]} L_c \geq 1$. By Lemma G.2, this point satisfies Eq.43. Hence, this point satisfies Eq.45 if the convex hull $\{w_c\}_{c \in C}$ forms a regular $(C-1)$-simplex. Since $Z_c = 1$ for any $c \in [C]$ by Lemma G.2 and $Z_c \geq L_c \geq 1$, $L_c = Z_c = 1$. This implies that the bary-center $w'_c(\bar{\alpha})$ is the point in $\Delta_c$ closest to $w_c$. In the following, we argue that this property implies that $\|w_{c_1} - w_{c_2}\|$ is independent of the choice of distinct pair $c_1, c_2 \in [C]$, implying that the convex hull $\{w_c\}_{c \in [C]}$ forms a regular simplex.

For any $c_1 \in [C]$, $\|w'_{c_1}(\bar{\alpha})\| = \|w_{c_1}\| / (C-1) = C^{-1}$. Since $L_c > 0$ for any $c \in [C]$, all the vector $w_c$ are distinct, and hence, $w'_{c_1}(\bar{\alpha})$ is perpendicular to $w_{c_2} - w'_{c_1}(\bar{\alpha})$. Therefore,

$$\left\| w_{c_2} - w'_{c_1}(\bar{\alpha}) \right\|^2 = \|w_{c_2}\|^2 - \left\| w'_{c_1}(\bar{\alpha}) \right\|^2 = \frac{(C-1)^2}{C^2} - C^{-2} = \frac{C-2}{C}.$$

Thus,

$$\begin{aligned}
\|w_{c_2} - w_{c_1}\|^2 &= \left\| w_{c_2} - w'_{c_1}(\bar{\alpha}) \right\|^2 + \left\| w_{c_1} - w'_{c_1}(\bar{\alpha}) \right\|^2 \\
&= (C-2)/C + 1 \\
&= 2C^{-1}(C-1).
\end{aligned}$$

This is independent of $c_1$ and $c_2$, implying that the simplex is regular. $\qquad\square$

**Lemma G.4.** *Under assumption A9 and $d \geq C$, the minimization Eq.35 is achieved if and only if the following conditions are met:*

$$\begin{cases}
\text{The convex hull of } \{w_c\}_{c \in [C]} \text{ forms a regular } (C-1)\text{-simplex} \\
\|w_c\| = \frac{C-1}{C} \quad \forall c \in [C] \\
\sum_{c \in [C]} w_c = 0 \\
b = 0 \\
h_i = \frac{C}{C-1} w_{y_i} \quad \forall i \in [n].
\end{cases} \tag{46}$$

*Proof.* We first show that a minimizer of Eq.35 satisfies Eq.46. For a minimizer $(\{w_c\}_{c \in [C]}, b, \{h_i\}_{i \in [n]})$ of Eq.35, $(\{w_c\}_{c \in [C]}, b)$ is a minimizer of Eq.36. Thus, by Lemma G.3, the minimizer satisfies the first four properties. Additionally, by Lemma G.1,

$$\max_{\{h_i\}} \left\{ \min_{i \in [n]} q_i : \sum_j (h_i)_j^2 = 1 \quad \forall i \in [C] \right\} = \min_{c \in [C]} L_c = 1.$$

Therefore, to satisfy the constraint $\min_{i \in [n]} q_i \geq 1$ in Eq.35, $\{h_i\}_{i \in [n]}$ needs to be a maximizer of the equation above. Again by Lemma G.1, this maximizer is given by

$$h_i = \frac{w_{y_i} - w^*_{y_i}}{\left\| w_{y_i} - w^*_{y_i} \right\|} = \frac{w_{y_i} - w'_{y_i}(\bar{\alpha})}{\left\| w_{y_i} - w'_{y_i}(\bar{\alpha}) \right\|} = \frac{C}{C-1} w_{y_i},$$

which is the last condition.

We now prove the inverse. We assume that $(\{w_c\}_{c \in [C]}, b, \{h_i\}_{i \in [n]})$ satisfies Eq.46. By Lemma G.3, $(\{w_c\}_{c \in [C]}, b)$ is a minimizer of Eq.36. Since the minimum value of Eq.36 is equivalent to the minimum value of Eq.35, it suffices to show that $(\{w_c\}_{c \in [C]}, b, \{h_i\}_{i \in [n]})$ satisfies the constraints in Eq.35, which can be seen as follows:

$$\min_{i \in [n]} q_i = \min_{i \in [n]} \min_{c \in [C]/\{y_i\}} (w_{y_i} - w_c)^T \frac{C}{C-1} w_{y_i} = \frac{C-1}{C} - \frac{-1}{C} = 1$$

$$\sum_j (h_i)_j^2 = \sum_j (\frac{C}{C-1} w_{y_i})_j^2 = 1.$$

$\qquad\square$

**Lemma G.5.** *Under assumption A9 and $d \geq C$, the minimization Eq.33 is achieved if and only if the following conditions are met:*

$$
\begin{cases}
\text{The convex hull of } \{w_c\}_{c \in [C]} \text{ forms a regular } (C-1)\text{-simplex} \\
\|w_c\| = \frac{C-1}{C} \quad \forall c \in [C] \\
\sum_{c \in [C]} w_c = 0 \\
b = 0 \\
h_i = \frac{C}{C-1} w_{y_i} \quad \forall i \in [n] \\
\sum_{i \in [d]} (w_c)_i = 0 \quad \forall c \in [C].
\end{cases}
\tag{47}
$$

*Proof.* Since Eq.33 has an additional constraint

$$
\sum_j (h_i)_j = 0 \quad \forall i \in [C],
\tag{48}
$$

compared to Eq.35, by Lemma G.4, it suffices to show

- $(\{w_c\}_{c \in [C]}, b, \{h_i\}_{i \in [n]})$ satisfies Eq.46 and Eq.48 if and only if Eq.47 is met.

- There exists $(\{w_c\}_{c \in [C]}, b, \{h_i\}_{i \in [n]})$ satisfying Eq.47.

First we show the first statement. We assume Eq.46 and Eq.48. Then the last equality in Eq.47 holds as follows:

$$
\sum_{j \in [d]} (w_c)_j = \sum_{j \in [d]} \left( \frac{C-1}{C} h_i \right)_j = 0,
$$

where $i \in [n]$ is such that $y_i = c$. The existence is assured by A9. The other equalities in Eq.47 trivially holds since they are equivalent to Eq.46. Conversely, if Eq.47 holds, Eq.46 trivially holds and Eq.48 holds as well since

$$
\sum_{j \in [d]} (h_i)_j = \sum_j \left( \frac{C}{C-1} w_{y_i} \right)_j = 0.
$$

The existence of $(\{w_c\}_{c \in [C]}, b, \{h_i\}_{i \in [n]})$ satisfying Eq.47 can be seen by the fact that regular $(C-1)$-simplex is in $(C-1)$-dimensional subspace, and we can rotate the simplex such that it is inside the $(d-1)$-dimensional subspace constrained by $\sum_{i \in [d]} (w_c)_i = 0 \quad \forall c \in [C]$, without violating the other equalities in Eq.47.

$\square$

*proof of Theorem G.1.* By A8, the optimization problem Eq.8 can be reduced to Eq.33. Hence, by Lemma G.5, the minimizer's condition is given by Eq.47, which implies the first three properties of Neural Collapse hold with $\bar{h}_c = h_c^*$. The last property can be proved as follows.

The distance between $h$ and $h_c^*$ is given by

$$
\begin{aligned}
\|h - h_c^*\|^2 &= \|h\|^2 + \|h_c^*\|^2 - 2h^T h_c^* \\
&= \|h\|^2 + 1 - \frac{2C}{C-1} h^T w_c.
\end{aligned}
$$

Hence,

$$
\begin{aligned}
\operatorname{argmin}_{c \in [C]} \|h - h_c^*\| &= \operatorname{argmin}_{c \in [C]} \left( \|h\|^2 + 1 - \frac{2C}{C-1} h^T w_c \right) \\
&= \operatorname{argmax}_{c \in [C]} h^T w_c \\
&= \operatorname{argmax}_{c \in [C]} h^T w_c + b_c.
\end{aligned}
$$

$\square$

## H   EXTENSION TO MULTI-CLASS CLASSIFICATION

In this section, we extend Theorem 4.1 to the case of multi-classification tasks with cross-entropy loss. The analysis here largely relies on Appendix G in Lyu & Li (2019).

We consider a classification task of data $\{x_i, y_i\}_{i \in [n]}$ whose label $y_i$ which now takes values in $[C]$, where $C \in \mathbb{N}$ is the number of classes. Our model's output is given by $f(x; \theta) \in \mathbb{R}^C$, and the cross-entropy loss with this model is defined as

$$\mathcal{L} := -n^{-1} \sum_{j \in [n]} \log \frac{\exp(f_{y_j}(x_j; \theta))}{\sum_{c \in [C]} \exp(f_c(x_j; \theta))} = n^{-1} \sum_{j \in [n]} \log(1 + e^{-\tilde{q}_j}), \tag{49}$$

where $\tilde{q}_j := -\log \left( \sum_{c \in [C]/\{y_j\}} e^{-s_{jc}} \right)$, and $s_{jc} := f_{y_j}(x_j : \theta) - f_c(x_j; \theta)$. Notice that $s_{jc}$ is a quasi-homogeneous function. Hence, if Lemma E.2 holds and $\theta$ goes to infinity as $t \to \infty$, which we will show later in this section, $s_{jc}$ goes to infinity. Therefore, when $t \gg 1$,

$$\tilde{q}_j \sim -\log \left( \max_{c \in [C]/\{y_j\}} e^{-s_{jc}} \right) = \min_{c \in [C]/\{y_j\}} s_{jc} \ (\to \infty),$$

and by Taylor expansion of logarithm $\log(1 + e^{-\tilde{q}_j}) \sim e^{-\tilde{q}_j}$,

$$\mathcal{L} = \sum_{j \in [n]} \log(1 + e^{-\tilde{q}_j}) \sim \sum_{j \in [n]} e^{-\tilde{q}_j}.$$

This expression is now equivalent to the one of binary classification tasks with the exponential loss. Note that the effective margin is now given by $\min_{c \in [C]/\{y_j\}} s_{jc}$, which implies that its asymptotic behavior at the later stage of training is similar to the one with the exponential loss. Being aware of this fact, we can show a variant of Theorem 4.1 with a modified version of separability condition:

**A10** (*Strong Separability for CE Loss*). There exists a time $t_0$ such that $\mathcal{L}(\theta(t_0)) < n^{-1} \log 2$.

Under A1,A2, A4, A6, A7, and , A10 with cross-entropy loss Eq.49, there exists an $\alpha \in \mathbb{R}$ such that $\psi_\alpha(\lim_{t \to \infty} \hat{\theta}(t))$ is a first-order KKT point of the following optimization problem

$$\begin{aligned} \text{minimize} \quad & \frac{1}{2} \|\theta\|^2_{\Lambda_{\max}} \\ \text{subject to} \quad & \min_{c \in [C]/\{y_j\}} s_{jc} \geq 1 \quad \forall j \in [n]. \end{aligned}$$

The modification of our proof of Theorem 4.1 is quite similar to the extension done in Lyu & Li (2019) and straightforward except the part where we show the lower bound of $\frac{d}{dt} \log \|\theta\|_\Lambda$ and divergence $\|\theta\|_\Lambda \to \infty$. Hence here we will focus on this non-trivial part and ask readers to refer Lyu & Li (2019) and Nacson et al. (2019b) for the other details.

Similar to the first inequality of Lemma 4.1, we can show the following:

$$\frac{1}{2} \frac{d}{dt} \|\theta\|^2_\Lambda \geq (1 - e^{-n\mathcal{L}}) \log \left( e^{n\mathcal{L}} - 1 \right). \tag{50}$$

It is easy to see that

$$\begin{aligned} \frac{1}{2} \frac{d}{dt} \|\theta\|^2_\Lambda &= \left\langle \frac{d\theta}{dt}, \Lambda\theta \right\rangle \\ &= \sum_{j \in [n]} \frac{1}{1 + e^{-\tilde{q}_j}} \sum_{c \in [C]/\{y_j\}} e^{-s_{jc}} \langle h_{jc}, \Lambda\theta \rangle \quad (h_{jc} \text{ is some element of } \partial^\circ_\theta s_{jc}) \\ &= \sum_{j \in [n]} \frac{1}{1 + e^{-\tilde{q}_j}} \sum_{c \in [C]/\{y_j\}} e^{-s_{jc}} s_{jc} \\ &\geq \sum_{j \in [n]} \frac{1}{1 + e^{-\tilde{q}_j}} \sum_{c \in [C]/\{y_j\}} e^{-s_{jc}} \tilde{q}_j \\ &= \sum_{j \in [n]} \frac{\tilde{q}_j e^{-\tilde{q}_j}}{1 + e^{-\tilde{q}_j}}. \end{aligned}$$

Since $\tilde{q}_j$ can be uniformly lower bounded by $\mathcal{L}$ as follows

$$\mathcal{L} \geq n^{-1}\log(1 + e^{-\tilde{q}_j}), \text{ i.e., } \tilde{q}_j \geq -\log\left(e^{n\mathcal{L}} - 1\right),$$

$$\sum_{j \in [n]} \frac{\tilde{q}_j e^{-\tilde{q}_j}}{(1 + e^{-\tilde{q}_j})\log(1 + e^{-\tilde{q}_j})} \geq \frac{-\log\left(e^{n\mathcal{L}} - 1\right)e^{\log\left(e^{n\mathcal{L}} - 1\right)}}{(1 + e^{\log(e^{n\mathcal{L}} - 1)})\log(1 + e^{\log(e^{n\mathcal{L}} - 1)})}$$

$$= \frac{-\left(e^{n\mathcal{L}} - 1\right)\log\left(e^{n\mathcal{L}} - 1\right)}{n\mathcal{L}e^{n\mathcal{L}}}.$$

Here for the inequality, we exploit the fact that function $\frac{xe^{-x}}{(1+e^{-x})\log(1+e^{-x})}$ is an increasing function. With the help of this inequality, we get

$$\frac{1}{2}\frac{d}{dt}\|\theta\|_\Lambda^2 = \sum_{j \in [n]} \log(1 + e^{-\tilde{q}_j})\frac{\tilde{q}_j e^{-\tilde{q}_j}}{(1 + e^{-\tilde{q}_j})\log(1 + e^{-\tilde{q}_j})}$$

$$\geq \frac{-\left(e^{n\mathcal{L}} - 1\right)\log\left(e^{n\mathcal{L}} - 1\right)}{n\mathcal{L}e^{n\mathcal{L}}}\sum_{j \in [n]}\log(1 + e^{-\tilde{q}_j})$$

$$= (1 - e^{-n\mathcal{L}})\log\left(e^{n\mathcal{L}} - 1\right).$$

Next, we show a variant of Lemma E.2. By utilizing the lower bound of $\frac{1}{2}\frac{d}{dt}\|\theta\|_\Lambda^2$ and introducing a newly defined smoothed normalized margin $\tilde{\gamma} := -\frac{e^{n\mathcal{L}}-1}{\|\theta\|_\Lambda^{\lambda_{\max}^{-1}}}$,

$$-\frac{d\mathcal{L}}{dt} \geq \lambda_{\max}^{-1}\frac{(1 - e^{-n\mathcal{L}})^2\log\left(e^{n\mathcal{L}} - 1\right)^2}{\|\theta\|_\Lambda^2}$$

$$= \lambda_{\max}^{-1}(e^{n\mathcal{L}} - 1)^2\log(e^{n\mathcal{L}} - 1)^{(2-2\lambda_{\max})}\left(-\frac{e^{n\mathcal{L}} - 1}{\|\theta\|_\Lambda^{\lambda_{\max}^{-1}}}\right)^{2\lambda_{\max}}$$

$$\geq \lambda_{\max}^{-1}(e^{n\mathcal{L}} - 1)^2\log(e^{n\mathcal{L}} - 1)^{(2-2\lambda_{\max})}\tilde{\gamma}(t_0)^{2\lambda_{\max}}.$$

where the last inequality relies on a version of Corollary D.1. Rearranging terms on both sides of the inequality gives,

$$-\frac{d\mathcal{L}}{dt}(e^{n\mathcal{L}} - 1)^{-2}\log(e^{n\mathcal{L}} - 1)^{-2(1-\lambda_{\max})} \geq \lambda_{\max}^{-1}\tilde{\gamma}(t_0)^{2\lambda_{\max}}.$$

Integration from $t_0$ to $t$, with the substitution $-n\frac{d\mathcal{L}}{dt}(e^{n\mathcal{L}} - 1)^{-2} = \frac{d(e^{n\mathcal{L}}-1)^{-1}}{dt}$, gives

$$n^{-1}\int_{(e^{n\mathcal{L}(t_0)}-1)^{-1}}^{(e^{n\mathcal{L}(t)}-1)^{-1}}(-\log z)^{-2(1-\lambda_{\max})}dz \geq \lambda_{\max}^{-1}\tilde{\gamma}(t_0)^{2\lambda_{\max}}(t - t_0).$$

The RHS diverges as $t \to \infty$, and the LHS as a function of $t$ is non-decreasing for $z < 1$, which is true for all $t \geq t_0$ by Lemma D.1 and A10. Thus we can conclude that $\mathcal{L}^{-1} \to \infty$. Exploiting this fact, we can easily show $\|\theta\|_\Lambda \to \infty$ as we discussed in the proof of Lemma E.2.

## I    EXPERIMENT DETAILS

All empirical figures in this work were generated by the attached notebook. Here we briefly summarize the experimental conditions used to generate these figures.

**Logistic Regression (Fig. 1).**

This plot was generated by sampling 100 sample from two Gaussian distributions $\mathcal{N}(\pm\mu, \sigma I)$ in $\mathbb{R}^2$ where $\mu = [1/\sqrt{2}, 1/\sqrt{2}]$ and $\sigma = 0.25$. The parameters for both the homogeneous and quasi-homogeneous model were initialized as standard random Gaussian vectors. The parameters were trained with full batch gradient descent with a learning rate $\eta = 0.5$ for $1e5$ steps. The maximum $\ell_2$-margin solution was computed using scikit-learn's SVM package Pedregosa et al. (2011).

**Ball Classification (Fig. 4).**

This plot was generated by sampling $1e4$ random samples from the surface of two balls $B(\pm\mu, r)$ in $\mathbb{R}^3$ for 100 linearly spaced radii $r \in [0, 1]$. The mean $\mu = [0.8660254, 0.4330127, 0.25]$ was chosen such that $\rho_\mu = 0.5$ and $\rho_{P^\perp\mu} = 0.25$. The quasi-homogeneous model was defined such that $D_1 = 1$, $D_2 = 5$, and $D_3 = 10$ leading to $\lambda$-values $\Lambda = [1.0.2, 0.1]$. The parameters for the homogeneous and quasi-homogeneous model were initialized with the all ones vector. Using these initializations and SciPy's initial value problem ODE solver Virtanen et al. (2020) we then simulated gradient flow until $T = 1e5$. The final value of the classifier for both models and their respective robustness was recorded and used to generate the final plots.

