# OpenReview forum: "The Asymmetric Maximum Margin Bias of Quasi-Homogeneous Neural Networks"
_ICLR.cc/2023/Conference — ICLR 2023 notable top 25%_

### Official Review · Reviewer_UHdE · 2022-10-22

**Confidence:** 3
**Clarity, Quality, Novelty And Reproducibility:** The paper is novel and well-written.
**Correctness:** 4
**Technical Novelty And Significance:** 4
**Empirical Novelty And Significance:** Not applicable
**Recommendation:** 8

**Strength And Weaknesses:**

I think this is a nice paper. The quasi-homogeneous model is very expressive and allows common layers used in deep learning, which can be an interesting model to analyze. The proof naturally generalizes prior analysis on homogeneous networks, and reveals that gradient flow will asymptotically focus on the parameters with largest homogeneity factors. I have the following comments:
1. Part of Assumption A2 can be included in the definition of quasi-homogeneity, otherwise the definition may look too flexible. For example, given a deep linear network with no bias, we can let lambda be 1 for the last layer and 0 for other layers; this does not violate the definition, but the boundedness condition in A2 does not hold.
2. Since some $\lambda_i$ can be 0, $\hat{\theta}$ may not be uniquely defined. Can you make its definition more rigorous?
3. Can you explain how conditional separation (Assumption A7) is used in the proof?

**Summary Of The Paper:**

This paper analyzes the maximum-margin bias of gradient flow on quasi-homogeneous networks under data separation. In contrast to homogeneous networks, different parameters in a quasi-homogeneous network are allowed to have different homogeneity factors; this model can represent bias terms, skip connections, normalization layers, etc. This paper shows that the final maximum-margin bias is determined by the parameters with the maximum homogeneity factor, which naturally generalizes prior results on homogeneous networks. Specifically, it is proved that the final solution is a KKT point of a norm-minimization problem for the parameters with the maximum homogeneity factor, under a data separation condition. It is also discussed that quasi-homogeneous maximum margin can degrade robustness and induce neural collapse.

**Summary Of The Review:**

This paper introduces an expressive model of quasi-homogeneous networks, and analyzes the maximum-margin bias of gradient flow on such models.

---

> ### Author Response · Authors · 2022-11-11
> **Response to Reviewer UHdE**
>
> Thank you for your positive review.  Please see our updated manuscript which has a more detailed discussion on our assumptions and the choice of $\Lambda$.  Here we address your three comments individually:
>
> 1. You are correct, the boundedness condition in A2 was too strict. We can actually remove this aspect of A2 entirely because A6 (the convergence of the normalized parameters) is sufficient to show the boundedness of the function along the normalized training trajectory (new Lemma E.1).
> 2. As long as the $\Lambda$-seminorm is non-zero (i.e. $\|\theta\|_\Lambda > 0$),  $\hat{\theta}$ is uniquely defined. This is true for all positive semi-definite $\Lambda$.  Please see new Lemma A.1 for a formal statement and proof.
> 3. Assumption 7 is used to show that the normalized margin is upper-bounded (see Lemma E.2 for the technical details).  Intuitively, A7 is needed to guarantee that $\tilde{\gamma}$ converges.  For a homogeneous model, A7 is trivially satisfied and similarly, it is easy to show that $\tilde{\gamma}$ is bounded because it is invariant under scaling.  However, for a general $\Lambda$-quasi-homogeneous setting, $\tilde{\gamma}$ is not invariant, and thus we must introduce A7 to guarantee it does not diverge. Please see our new Appendix section C for a more detailed discussion on the limitations of our assumptions.
>
> Please let us know if you have any further questions or comments.  Thank you again for the time you spent reviewing our work.

---

### Official Review · Reviewer_i1KS · 2022-10-22

**Confidence:** 4
**Correctness:** 4
**Technical Novelty And Significance:** 4
**Empirical Novelty And Significance:** 3
**Recommendation:** 10

**Clarity, Quality, Novelty And Reproducibility:**

The paper is well-written, the results are novel and the experiments are well-documented.

**Strength And Weaknesses:**

The implicit bias of gradient flow in neural networks has been extensively studied in recent years. It is believed that understanding the implicit bias may be a key to understanding generalization in deep learning. In the context of classification with exponential-type losses, the most prominent prior result is by Lyu and Li (2019), and it implies a bias towards margin maximization in parameter space (or at least towards stationary points of the margin maximization problem) in homogeneous networks. The current work extends this result to a large family of non-homogenous networks, which includes ResNets and networks with bias terms. I believe that this contribution is very significant, as it makes a step towards understanding the implicit bias in practical settings. Therefore, I think that it is a very strong paper.

My main comment is that I think that the paper can benefit from a more detailed discussion on the assumptions made in Theorem 4.1. The authors give several examples of quasi-homogeneous networks, both in the main text and in the appendix, and convince the reader that the class of quasi-homogeneous networks is very rich and includes “nearly all neural networks with homogeneous activations”. While I agree with this statement, I think that the more relevant question is when the other assumptions hold. For example, the authors discuss “networks with residual connections” as an example of quasi-homogeneous networks. However, it seems that these networks do not satisfy Assumption A2, since f is not bounded on the unit “sphere” (in this example, the lambda of the first layer is 0 and of the second layer is positive, hence the unit sphere includes all networks where the second layer’s weight is 1, and by increasing the first layer we can get arbitrarily large outputs). I suspect that many readers might get the impression that since the family of quasi-homogeneous networks is very rich, then the theorem essentially holds in all interesting settings, but it does not seem to be the case. Therefore, such a discussion, in my opinion, is important. A more concrete question for discussion is: when does the theorem apply in ResNets? (I suppose that if the residual block includes only trainable parameters, both skipping and non-skipping, and does not include the identity matrix, then the assumptions should hold if the skip-connections are necessary for separating the data and assuming directional convergence, right?)

Some additional comments:
- In the proof of Theorem 4.1, you show that the KKT point satisfies the MFCQ condition. It implies that the KKT conditions are necessary for optimality, right? I think that it is a useful fact and it might be worth mentioning in the main text.
- Page 6: “See App. C for a discussion on how to determine the highest-order…” — Do you mean App. A?
- Page 7: “$(w_3)$” should be “$(w_1)$”?
- In Sec. 5, it might be worth discussing that also homogeneous linear networks may converge to non-robust solutions. For example, diagonal linear networks prefer sparse solutions and do not maximize the $\ell_2$ margin.



**Summary Of The Paper:**

The paper studies the implicit bias of gradient flow with an exponential loss on a class of neural networks called “quasi-homogeneous networks”. This class, which is introduced in the paper, is very rich and includes, e.g., ReLU networks with bias terms and residual connections. The implicit bias of such networks is not implied by previous results, which consider mostly homogeneous networks. The results hold for both binary classification, and multiclass classification with the cross-entropy loss. The implicit bias of quasi-homogeneous networks is characterized by asymmetric margin maximization (which involves only a subset of the parameters). The authors also conjecture that an even more general property holds (a certain cascading weight minimization property). Also, they obtain some implications in the context of robustness and neural collapse.

**Summary Of The Review:**

As I discussed above, I think that it is a strong paper and recommend “strong accept”.

-----------------------------------------------

After reading the authors' response, I stick with my high score.

---

> ### Author Response · Authors · 2022-11-11
> **Response to Reviewer i1KS**
>
> Thank you for your very positive review!  Please see our updated manuscript which has a more detailed discussion on our assumptions and the choice of $\Lambda$.
>
> We completely agree with your comment that “the paper can benefit from a more detailed discussion on the assumptions made in Theorem 4.1”.  We have decided to add a whole section to the appendix discussing these assumptions and their limitations. Please see the new App. C section.
>
> Thank you for pointing out the issue between A2 and networks with residual connections.  We realized that A2 was too strong and as also pointed out by Reviewer UHdE.  We have removed the boundedness condition of A2 entirely because A6 (the convergence of the normalized parameters) is sufficient to show the boundedness of the function along the normalized training trajectory (new Lemma E.1).  In short, Theorem 4.1 will apply to all ResNet-18 models if A6 is satisfied.
>
> In cases where the residual block diverges, the normalized parameter $\hat \theta$ does not converge, and hence Theorem 4.1, which is about the converged value of $\hat \theta$, does not hold. Additionally, the divergence of the residual block means that the skip connection part is effectively negligible, and does not play an important role. Hence, we believe that if the skip connections play an important role in the classification task, the residual block does not diverge.
>
> Here we address your additional comments individually:
> - Yes, this is clearly an important point that we forget to mention. We added a new separated lemma (Lemma A.4) saying any feasible point satisfies MFCQ and added a footnote in the main text of Theorem 4.1.
> - Yes, we fixed this reference.
> - Yes, this was a typo.
> - We added a footnote to section 5 mentioning that for higher-order homogeneous models such as a deep linear network the maximum margin bias towards sparse solutions could erode the robustness.
>
> Please let us know if you have any further questions or comments.  Thank you again for the time you spent reviewing our work.

---

> > ### Comment · Reviewer_i1KS · 2022-11-13
> > **A question**
> >
> > Thanks for your response.
> > I still don't understand something regarding Assumption A6.
> > You say that "A6 essentially assumes the convergence of the decision boundary" and "A6 is necessary for a technical reason, but we expect that this assumption can be weakened or removed...". However, you also say that it does not hold in the ResNet example if the residual block diverges. If I understand correctly, there are two contradictions here:
> >
> > (1) It is possible that the residual block diverges, but the decision boundary converges. For example, it can happen in ResNets in the simple case of a linear activation function and input dimension 1.
> >
> > (2) If we remove Assumption A6, then the ResNet example will satisfy all the assumptions, but the theorem statement will not hold since $lim_{t \to \infty} \hat{\theta}$ does not even exist.

---

> > > ### Author Response · Authors · 2022-11-14
> > > **Answer to Reviewer i1KS's Question**
> > >
> > > We appreciate your careful reading! Let us clarify the two sentences you pointed out.
> > >
> > > (1) You are correct, the sentence “A6 essentially assumes the convergence of the decision boundary” is misleading, in the sense that the two conditions are NOT necessarily equivalent. While A6 **implies** the convergence of the decision boundary, the convergence of the decision boundary does not imply A6, such as in the example you pointed out. In other words, A6 is a stronger condition than the convergence of decision boundary in some cases. This is because, in some cases, there can be multiple points representing the same decision boundary on the unit $\Lambda$-seminorm space. Hence, the convergence of the decision boundary does not necessarily mean that $\hat \theta$ converges. We will change our sentence to make this point clear.
> > >
> > > (2) You are correct, it is impossible to completely remove A6 for free, but it might be possible to make it weaker. However, it is currently unknown to us how much we can relax this condition. Lyu & Li are able to avoid A6 for a homogeneous model through a sophisticated argument of the "limit points" of $\hat{\theta}$, but this argument cannot be applied to the general quasi-homogeneous case directly. We will edit our discussion to make it clear that A6 cannot be removed completely.
> > >
> > > Thank you for your great question.

---

> > > > ### Comment · Reviewer_i1KS · 2022-11-15
> > > > **👍**
> > > >
> > > > Thanks

---

### Official Review · Reviewer_CPzm · 2022-10-23

**Confidence:** 3
**Correctness:** 2
**Technical Novelty And Significance:** 4
**Empirical Novelty And Significance:** 3
**Recommendation:** 8

**Clarity, Quality, Novelty And Reproducibility:**

Quality. Most of the theoretical results are strong and solid, except the inconsistency of Theorem 4.1.

Novelty. The definition of quasi-homogeneous networks and the results on their inductive bias are new.

Clarity. Mostly the paper is well-written.


**Strength And Weaknesses:**

Strengths:
1. Strong and novel theoretical results.
2. Novel definition of quasi-homogeneous networks that may help to understand neural networks in new settings.
3. Interesting experiments on robustness.

Weaknesses:

The major weakness is that Theorem 4.1 in its current form is inconsistent. Since the quasi-homogeneous parameter is not unique, for a given network the theorem implies that gradient flow converges to different minimum norm problems. This is impossible, unless:
1. All such problems are equivalent, or:
2. Given the assumptions, there is a unique parameterization.

The authors provide several examples where (1) or (2) above hold, but there is no general proof that they hold for all quasi-homogeneous models. If the theorem implies that either (1) or (2) hold for all quasi-homogeneous models, then this should be stated in the theorem.

Another issue with the inconsistency is that Definition 3.1 holds for all x. However, the main result only concerns with the training set, therefore it seems that the definition can be only with respect to x in the training set. If we change definition 3.1 to hold for x in the training set, this changes the possible parameterizations of quasi-homogeneous models and the main result. It is not clear which definition should be used.


Another weakness is that conditional separation assumption seems strong. There should be more discussion on this assumption and maybe proofs that it holds in certain realistic settings.


**Summary Of The Paper:**

This work studies the implicit bias of quasi-homogeneous networks. Under a conditional separation assumption, it is proved that gradient flow converges to a KKT solution of a certain minimum norm optimization problem. The latter norm depends on a subset of the parameters such as the last layer bias of a fully connected network. Based on the main result, it is shown in a certain setting that the inductive bias of quasi-homogeneous networks is towards non-robust solutions. Furthermore, it is proved that the neural collapse phenomenon occurs for quasi-homogeneous models under certain assumptions.

**Summary Of The Review:**

This paper presents new and interesting results on the inductive bias of non-homogeneous networks. However, I cannot accept this paper in its current form, because one of the theoretical results is currently inconsistent. If the authors can clarify the statements and write a consistent theorem then I will increase the score.

---

> ### Author Response · Authors · 2022-11-11
> **Response to Reviewer CPzm**
>
> Thank you for your review and the time you spent suggesting ways to make our work better. Let us discuss the weaknesses in our paper you pointed out:
>
> **Non-uniqueness of $\Lambda$:**
> We believe that the proof of Theorem 4.1 itself ensures the consistency of the theorem, but as you pointed out, it is not very obvious that your bullet points (1) or (2) holds, and hence the theorem looks inconsistent. To further clarify this point, we generalized our previous example-based argument in the Appendix, and wrote down a proof for the general setting, showing that your bullet points (1) or (2) indeed hold. (Please see Appendix B)
> In particular, we consider the set of $\Lambda$ with which the model is quasi-homogeneous, and we show:
> 1. The proper choice of $\Lambda$ satisfying assumptions is unique in the interior of the set, i.e., your bullet point (2).
> 2. Although there can be multiple proper choices of $\Lambda$ on the boundary of the set, the KKT conditions derived from those $\Lambda$ are unique, i.e., your bullet point (1).
>
> **Inconsistency in Definition 3.1:**
> Definition 3.1 is independent of the data, and it is solely a property of the model architecture. We believe that your point is that definition 3.1 could be relaxed to quasi-homogeneity only for the given training dataset, and then Theorem 4.1 would still hold.  This is likely true, but there is no inconsistency by using our more strict, data-independent definition 3.1.
>
> **A7 is strong:**
> We agree, A7 is strong, which is why we considered what happens without it and introduced Conjecture 5.1. While strong, A7 is actually true for many practical networks such as ResNet-18 with batch normalization.  In this case, the last layer parameters, which are $\Lambda_{\max}$ parameters, are necessary for classification and thus this network will always satisfy A7.  This is what allows us to apply Theorem 4.1 in section 6 and reveal a mechanism behind Neural Collapse.  Please see our new Appendix section C where we have a more detailed discussion on the limitations of our assumptions and a possible route towards proving Conjecture 5.1.
>
>
> Thank you again for the time you spent reviewing our work and we hope you will consider raising your score. Please let us know if you have any further questions or comments.

---

> > ### Comment · Reviewer_CPzm · 2022-11-18
> > **Thanks for the response**
> >
> > I did not go over the math details of Appendix B, but the theorem statement looks ok to me. The other points I mentioned were addressed. I raised the score.

---

### Official Review · Reviewer_bFVN · 2022-10-27

**Confidence:** 3
**Correctness:** 3
**Technical Novelty And Significance:** 4
**Empirical Novelty And Significance:** Not applicable
**Recommendation:** 8

**Clarity, Quality, Novelty And Reproducibility:**

The writing is mostly clear. The paper is well-written and I found it easy to digest.

Analyzing the class of quasi-homogeneous networks looks novel, as far as I can tell. In terms of proof techniques, the paper indeed builds upon the techniques developed in Lyu and Li (2019) but it looks like the authors had to overcome some challenges that arise from quasi-homogeneity. However, I did not have the time to fully examine the proofs.


**Strength And Weaknesses:**

Strengths

1. This paper generalizes the existing theoretical research on homogeneous networks to a greater extent, and the newly introduced class of quasi-homogeneous networks is very general. Hence, the paper develops a theory that applies to many practical networks.

2. Through this extension, the paper uncovers that the max margin bias of gradient flow only applies with respect to a subset of parameters, which is quite an interesting discovery. Moreover, this observation can help explain the empirically observed neural collapse phenomenon.

3. Although I believe assumption A7 is unrealistic in some practical scenarios (see the weaknesses part), the authors present a motivating example (Section 5) that allows us to imagine what might happen without the assumption. They propose a neat conjecture (Conjecture 5.1) which naturally extends Theorem 4.1 to settings without A7. This uncovers an interesting future research direction.

Weaknesses

4. Among the assumptions required for Theorem 4.1, the one that seemed the most unrealistic to me was A7, namely that the highest-degree parameters must contribute to the classifier for it to separate the dataset. The assumption does not hold in some practical scenarios, for example when we have a 1-hidden-layer fully-connected network with bias terms (hence the bias $b^2$ of the output layer is the highest-degree parameter) and the dataset can be separated without the use of $b^2$.

5. As noted by the authors, the diagonal PSD matrix $\Lambda$ that defines quasi-homogeneity may not be unique. For example, for networks with normalization layers, one can choose *any* values of $\lambda$ for scale-invariant parameters. The authors discuss in Appendix A on how we can disambiguate between different possible choices of $\Lambda$ by leveraging assumption A7 and taking a closer look at the optimization problem (P). However, this is only a "proof by example" and the paper does not concretely prove that there is only one possible choice of $\Lambda$ for which Theorem 4.1 applies. Also, it is unclear how we can disambiguate the equivalently working values of $\Lambda$ without the help of A7. I believe this is the main downside of this paper.

Minor comments
- To me, the word "highest-order" for parameters with $\lambda_{\max}$ seems a bit counterintuitive, because these parameters are often those with the lowest "polynomial degree." For example, for $f(x;\theta) = w^2 \sigma(w^1 x + b^1) + b^2$, the $b^2$ term with the smallest "degree."
- In "Geometric properties" paragraph on page 4, $\nabla_\alpha f(x; \psi_\alpha(\theta)) = \nabla_\alpha e^\alpha f(x;\theta)$?
- From Lyu and Li (2019) I believe A2 is stated for a fixed given value of $x$, but the paper doesn't clearly state that. As a result, I once got confused why $|f(x;\theta)| \leq k$ must hold for all $x$.
- It'd be good to put a label (P) next to the optimization problem in Theorem 4.1?
- At the end of Section 4, "See App C" -> "See App A".


**Summary Of The Paper:**

This paper studies the max margin bias that arise in the gradient flow training procedure of *quasi-homogeneous* neural networks. Quasi-homogeneous networks can be thought of as a generalization of homogeneous networks that are previously studied in the literature (e.g., Lyu & Li (2019) and Ji & Telgarsky (2020)), and they contain a significantly wider range of networks such as ones with bias terms, skip connections, normalization layers, etc.

For this general class of networks, the paper extends the results by Lyu & Li (2019) to show that the direction at which the network parameter vector converges to corresponds to a first-order KKT point of a (nonconvex) optimization problem (denoted as (P) in the paper). The key difference from Lyu & Li (2019) is that the objective function in (P) is no longer the standard $\ell_2$ norm; instead, it is the $\ell_2$ norm of a *subset* of parameters defined by the quasi-homogeneity.

This result highlights that the gradient flow dynamics is biased toward achieving the maximum margin with respect to only a subset of network parameters. This "asymmetry" leads to some interesting consequences, namely that 1) it can hurt robustness of the resulting classifier and 2) it can provide an explanation to the recently observed *neural collapse* phenomenon.

**Summary Of The Review:**

The paper investigates the max margin bias in a very general class of neural networks referred to as quasi-homogeneous networks. The investigation uncovers some interesting and unexpected bias, namely the implicit bias applies with respect to only a subset of parameters. Also, this theoretical result has some important implications to neural collapsing which is of interest.

While the paper presents interesting theoretical results, one crucial downside is that the quasi-homogeneity is not completely well-defined. For a given network there can be multiple "subsets" of network parameters, and fully disambiguating these cases could use some additional effort.

Overall, my current rating is positive but not too positive; I would be happy to raise my score based on the authors' response.

---

> ### Author Response · Authors · 2022-11-11
> **Response to Reviewer bFVN**
>
> Thank you for your positive review and the time you spent pointing out ways to make our work better. First, let us discuss the weaknesses in our paper you pointed out:
>
> **A7 is strong:**
> We agree, A7 is our strongest assumption, but necessary in our proof.  That said, we did consider what happens when this assumption does not hold.  That is why we introduced Conjecture 5.1 and provided empirical evidence supporting its claim.  We think further generalizing our proof of Theorem 4.1 to Conjecture 5.1 is an interesting direction for future work.  Please see our new Appendix section C where we have a more detailed discussion on the limitations of our assumptions and a possible route towards proving Conjecture 5.1.  Also, for your example of a 1-hidden-layer fully-connected network with bias terms, you are correct that if the dataset can be separated without $b^2$, then this setting will not satisfy A7.  That said, there can be datasets where $b^2$ is necessary, for example, for a network with ReLU activations trained on the binary classification task of separating two concentric spheres, the second layer bias can be necessary.  Thus, the validity of A7 will depend on the interaction of the parameterization of a model and the dataset, and thus it is difficult to assert its validity for general settings, without directly modeling the data as in section 5.
>
> **Non-uniqueness of $\Lambda$:**
> The non-uniqueness of $\Lambda$ is not an issue on its own because $\Lambda$-Quasi-homogeneity is a property of a function.  Definition 3.1 is well defined, even if there exist multiple valid values of $\Lambda$ which satisfy it.  Of course, as you mentioned, sometimes it is useful to distinguish among these valid values of $\Lambda$ such that there is no ambiguity when discussing for example the set of $\Lambda_{\max}$ parameters such as in Theorem 4.1.  In our new appendix B section, we thoroughly expand our previous “proof by example” argument and rigorously show that if the assumptions of Theorem 4.1 are met, then the resulting optimization problem (P) is unique.  The key to our proof strategy is defining a “proper” $\Lambda$ and showing that it is unique in the interior of possible $\Lambda$ values and that on the boundary it is possible to have multiple proper $\Lambda$ values, but their resulting optimization problems are equivalent. Please see our new App. B where we present the definition of a proper $\Lambda$ and a rigorous proof of the consistency of Theorem 4.1.
>
> Here we address your minor comments individually:
> - Your point makes sense, we have replaced “higher-order” and “lower-order” with “higher-rate” and “lower-rate” respectively.
> - This is not a typo, but we did skip the step $\nabla_\alpha e^\alpha f(x;\theta) = e^\alpha f(x;\theta)$.  To prevent confusion we removed the equality and referenced the appendix where there is a complete derivation.
> - You are correct, we added “For any fixed $x$,” to the beginning of Assumption 2.
> - We added the label (P), thanks for the suggestion.
> - Yes, this is a typo.
>
> Thank you again for the time you spent reviewing our work and we hope you will consider raising your score. Please let us know if you have any further questions or comments.

---

> > ### Comment · Reviewer_bFVN · 2022-12-02
> > **Response Acknowledged**
> >
> > Thanks for the response! I didn't have a chance to read through all the details of Appendix B, but it seems that the authors address the ambiguity issue that I raised; along with other positive reviews, I'm also happy to raise my score. I hope this paper will spur other follow-up results, including the removal of A7.

---

### Author Response · Authors · 2022-11-07
**Thank you to all reviewers.**

We would like to thank all reviewers for their careful and detailed comments.

We are happy to see that there is consensus among the reviewers on the significance and novelty of our contribution.  We also appreciate the common concern among reviewers around the strength of Assumption 7 and the non-uniqueness of $\Lambda$ in Definition 3.1. We are currently working on clarifying and explaining these aspects of our work and will provide an updated manuscript and individual replies by the end of the week.

We greatly appreciate the time and effort you spent reviewing our paper, which we are certain will make our work better.

---

### Author Response · Authors · 2022-11-11
**Updated Manuscript.**

Thank you again to all reviewers for the time and effort you spent reviewing our paper.  We have tried to incorporate all your feedback and suggestions in our updated manuscript.  Here is a list of the changes:

- Added App. B section where we define a proper $\Lambda$ and prove the consistency of Theorem 4.1 as suggested by reviewers bFVN and CPzm.
- Added App. C section discussing the intuition and limitations of our assumptions A1-A7 as suggested by reviewers CPzm and i1KS.
- We removed the boundedness condition of A2 and introduced Lemma E.1 proving boundedness of the normalized trajectory as suggested by reviewers i1KS and UHdE.
- Added Lemma A.1 proving the uniqueness of $\hat{\theta}$ whenever $\|\theta\|_\Lambda > 0$ as suggested by reviewer UHdE.
- Added figure 5 depicting the geometric intuition behind our proof of Neural Collapse in section 6.
- Added many smaller edits throughout the paper that are described in our individual responses to the reviewers.

Thank you again and we hope you will consider our paper an important contribution to the ICLR community.

---

### Decision · Program_Chairs · 2023-01-20

**Decision:**

Accept: notable-top-25%

**Justification For Why Not Higher Score:**

Although this is a clear acceptance, the main result of the paper is mostly an extension of known results in the literature, which explain why I do not recommend an oral presentation.

**Justification For Why Not Lower Score:**

N/A

**Metareview: Summary, Strengths And Weaknesses:**

This paper studies the max-margin bias that arises in the gradient flow training procedure of quasi-homogeneous neural networks. The latter includes a broad set of networks that are not homogeneous, therefore extending the current theory derived in prior work. The main result of the paper shows that GF converges to a first-order KKT point of a nonconvex optimization problem that minimizes a semi-norm defined in the paper.

This is an interesting and novel result. All reviewers agree this paper is worth publishing. Some minor issues were initially raised and addressed by the authors. This is a clear acceptance.

**Note From Pc:**

if the above contains the word "oral" or "spotlight" please see: "oral" presentation means -> notable-top-5% and "spotlight" means -> notable-top-25%. As stated in our emails, we are disassociating presentation type from AC recommendations

**Summary Of Ac-Reviewer Meeting:**

N/A